# On Local Equilibrium in Non-Concave Games

## Abstract

While Online Gradient Descent and other no-regret learning procedures are known to efficiently converge to coarse correlated equilibrium in games where each agent's utility is concave in their own strategies, this is not the case when the utilities are non-concave, a situation that is common in machine learning applications where the agents' strategies are parameterized by deep neural networks, or the agents' utilities are computed by a neural network, or both. Indeed, non-concave games present a host of game-theoretic and optimization challenges: (i) Nash equilibria may fail to exist; (ii) local Nash equilibria exist but are intractable; and (iii) mixed Nash, correlated, and coarse correlated equilibria have infinite support, in general, and are intractable. To sidestep these challenges we propose a new solution concept, termed *local correlated equilibrium*, which generalizes local Nash equilibrium. Importantly, we show that this solution concept captures the convergence guarantees of Online Gradient Descent and no-regret learning, which we show efficiently converge to this type of equilibrium in non-concave games with smooth utilities.

## 1 Introduction

Von Neumann's celebrated minimax theorem establishes the existence of Nash equilibrium in all two-player zero-sum games where the players' utilities are continuous as well as *concave* in their own strategy [Neu28].[1] This assumption that players' utilities are concave, or quasi-concave, in their own strategies has been cornerstone for the development of equilibrium theory in Economics, Game Theory, and a host of other theoretical and applied fields that make use of equilibrium concepts. In particular, (quasi-)concavity is key for showing the existence of many types of equilibrium, from generalizations of min-max equilibrium [Fan53; Sio58] to competitive equilibrium in exchange economies [AD54; McK54], mixed Nash equilibrium in finite normal-form games [Nas50], and, more generally, Nash equilibrium in (quasi-)concave games [Deb52; Ros65].

Not only are equilibria guaranteed to exist in concave games, but it is also well-established—thanks to a long line of work at the interface of game theory, learning and optimization whose origins can be traced to Dantzig's work on linear programming [Geo63], Brown and Robinson's work on fictitious play [Bro51; Rob51], Blackwell's approachability theorem [Bla56] and Hannan's consistency theory [Han57]—that several solution concepts are efficiently computable both centrally and via decentralized learning dynamics. For instance, it is well-known that the learning dynamics produced when the players of a game iteratively update their strategies using no-regret learning algorithms, such as online gradient descent, is guaranteed to converge to Nash equilibrium in two-player zero-sum concave games, and to coarse correlated equilibrium in multi-player general-sum concave games [CL06]. The existence of such simple decentralized dynamics further justifies using these solution concepts to predict the outcome of real-life multi-agent interactions where agents deploy strategies, obtain feedback, and use that feedback to update their strategies.

While (quasi)-concave utilities have been instrumental in the development of equilibrium theory, as described above, they are also too restrictive an assumption. Several modern applications and outstanding challenges in Machine Learning, from training Generative Adversarial Networks (GANs)

---

[1]Throughout this paper, we model games using the standard convention in Game Theory that each player has a utility function that they want to maximize. This is, of course, equivalent to modeling the players as loss minimizers, a modeling convention that is more common in learning. When we say that a player's utility is concave (respectively non-concave) in their strategy, this is the same as saying that the player's loss is convex (respectively non-convex) in their strategy.

to Multi-Agent Reinforcement Learning (MARL) as well as generic multi-agent Deep Learning settings where the agents' strategies are parameterized by deep neural networks or their utilities are computed by deep neural networks, or both, give rise to games where the agents' utilities are *non-concave* in their own strategies. We call these games *non-concave*, following [Das22].

Unfortunately, classical equilibrium theory quickly hits a wall in non-concave games. First, Nash equilibria are no longer guaranteed to exist. Second, while mixed Nash, correlated and coarse correlated equilibria do exist—under convexity and compactness of the strategy sets [Gli52], which we have been assuming all along in our discussion, they have infinite support, in general, and they are computationally intractable; so, a fortiori, they are also intractable to attain via decentralized learning dynamics. Finally, unlike non-convex optimization, where targeting local optima sidesteps the intractability of global optima, a natural multi-agent generalization of local optimum, called *local Nash equilibrium*—see Definition 1, has been recently shown to be intractable, even in two-player zero-sum non-concave games [DSZ21]. More broadly, the study of local equilibrium concepts that are guaranteed to exist in non-concave games has received a lot of attention in recent years—see e.g. [RBS16; HSZ17; DP18; JNJ20; DSZ21]. However, in terms of computing the local equilibrium concepts that have been proposed, existing results are restricted to sequential two-player zero-sum games [MV21]; or only establish local convergence guarantees for learning dynamics—see e.g. [DP18; WZB20; FCR20]; or only establish asymptotic convergence guarantees—see e.g. [Das+23]; or involve solution concepts that are non-standard in that their local stability is not with respect to a distribution over strategy profiles [HSZ17]. In view of the importance of non-concave games in emerging ML applications and the afore-described state-of-affairs, our investigation in this paper is motivated by the following, broad and largely open question:

**Question from [Das22]:** *Is there a theory of non-concave games? What solution concepts are meaningful, universal, and tractable?*

## 1.1 CONTRIBUTIONS

In this paper, we answer the question raised by [Das22] by proposing a new, general, local equilibrium concept, as well as two concrete instantiations of this concept which both are game-theoretically meaningful, universal, and computationally tractable. Importantly, we show that simple decentralized learning dynamics, e.g. the dynamics induced when each player updates their strategy using online gradient descent (GD), efficiently converges to our equilibrium concepts. Throughout the paper, we focus on differentiable games whose strategy sets are subsets of $\mathbb{R}^d$ and have $G$-Lipschitz and $L$-smooth (but not necessarily concave) utility functions (Assumption 1). Our contributions are as follows.

**Local Correlated Equilibrium.** A common way to sidestep the computational intractability of Nash equilibrium [DGP09] is to introduce *correlation* among the agents' strategies. Our local equilibrium concept uses this approach. It is a joint distribution over $\Pi_{i=1}^n \mathcal{X}_i$, the Cartesian product of all players' strategy sets, and is defined in terms of a set, $\Phi^{\mathcal{X}_i}(\delta)$, of $\delta$-*local strategy modifications*, for each player $i$. The set $\Phi^{\mathcal{X}_i}(\delta)$ contains functions mapping $\mathcal{X}_i$ to itself, and it satisfies that, for all $\phi_i \in \Phi^{\mathcal{X}_i}(\delta)$ and all $x \in \mathcal{X}_i$: $\|\phi_i(x) - x\| \leq \delta$. In terms of $\Phi(\delta) = \Pi_{i=1}^n \Phi^{\mathcal{X}_i}(\delta)$, we propose the notion of $(\varepsilon, \Phi(\delta))$-*local correlated equilibrium* to be a distribution over joint strategy profiles such that no player $i$ can increase their expected utility by more than $\varepsilon$ by applying any strategy modification function $\phi_i \in \Phi^{\mathcal{X}_i}(\delta)$ to the strategy sampled for them by the joint distribution. Local correlated equilibrium generalizes the notion of local Nash equilibrium, since any $(\varepsilon, \delta)$-local Nash equilibrium is, in fact, an $(\varepsilon, \Phi(\delta))$-local correlated equilibrium, for any choice of $\Phi(\delta)$. This also guarantees the existence of local correlated equilibrium in the regime where $\delta \leq \sqrt{2\varepsilon/L}$, which we refer to as the *local regime*, as $(\varepsilon, \delta)$-local Nash equilibria are guaranteed to exist in the same regime [DSZ21]. We instantiate our local correlated equilibrium concept by considering two natural choices for strategy modifications, $\Phi(\delta)$: (i) In the first instantiation, each player's set of local strategy modifications, denoted $\Phi^{\mathcal{X}}_{\text{Int}}(\delta)$, where $\mathcal{X}$ is $\mathcal{X}_i$ for player $i$, contains all deviations that *interpolate* between the input strategy and a fixed strategy, namely are of the form $\phi_{\lambda,x^*}(x) = (1-\lambda)x + \lambda x^*$, where $\lambda \leq \delta/D_{\mathcal{X}}$ and $D_{\mathcal{X}}$ is the diameter of $\mathcal{X}$. (ii) In the second instantiation, each player's set of local strategy modifications, denoted $\Phi^{\mathcal{X}}_{\text{Proj}}(\delta)$, contains all deviations that attempt a small step from their input in a fixed direction and project if necessary, namely are of the form $\phi_v(x) = \Pi_{\mathcal{X}}[x - v]$, where $\|v\| \leq \delta$ and $\Pi_{\mathcal{X}}$ stands for the $L_2$-projection onto $\mathcal{X}$.

**Local $\Phi$-Regret Minimization.** To efficiently compute a $(\varepsilon, \Phi(\delta))$-local correlated equilibrium in non-concave games, we draw a connection between online learning and game theory. We show that online learning algorithms that achieve low $\Phi^{\mathcal{X}}(\delta)$-regret against adversarial sequences of *convex* loss functions can be employed to converge to a local correlated equilibrium in a *non-concave* game, in the local regime of parameters (Lemma 1). While general $\Phi$-regret minimization algorithms result in sub-optimal guarantees and prohibitively high computational complexity (Section 3.1), we show that simple online learning algorithms such as Online Gradient Descent (GD) and Optimistic Gradient (OG) achieve optimal $\Phi^{\mathcal{X}}(\delta)$-regret for both $\Phi^{\mathcal{X}}_{\mathrm{Int}}(\delta)$ and $\Phi^{\mathcal{X}}_{\mathrm{Proj}}(\delta)$, namely:

- We show that $\Phi^{\mathcal{X}}_{\mathrm{Int}}(\delta)$-regret minimization reduces to the classical external regret minimization. Thus, the dynamics induced when each player runs GD or any no-regret learning algorithm efficiently converges to $\Phi^{\mathcal{X}}_{\mathrm{Int}}(\delta)$-local correlated equilibrium in the local regime. See Theorem 1.

- The notion of $\Phi^{\mathcal{X}}_{\mathrm{Proj}}(\delta)$-regret is incomparable to external regret (Examples 1 and 2). However, somewhat surprisingly, via a novel analysis we show that GD and OG achieve a near-optimal, $O(\sqrt{T})$, $\Phi^{\mathcal{X}}_{\mathrm{Proj}}(\delta)$-regret guarantee without any modification (Theorem 2 and Theorem 4). Moreover, when all players employ OG, each player enjoys an improved, $O(T^{1/4})$, $\Phi^{\mathcal{X}}_{\mathrm{Proj}}(\delta)$-regret (Theorem 5), breaking the $\Omega(\sqrt{T})$ lower bound in the adversarial setting (Theorem 3).

Our results complement existing results for learning in concave games. We establish that the notion of local correlated equilibrium characterizes the efficient convergence guarantees enjoyed by GD, OG, and other no-regret learning dynamics in non-concave games.

**Hardness in the Global Regime** A natural question that we haven't addressed yet is whether correlation is sufficiently powerful so that our solution concept becomes tractable even in the global regime of parameters (i.e. for large $\delta$). We provide a negative answer to this question by showing that when $\delta$ equals the diameter of our strategy set, it is NP-hard to compute an $(\varepsilon, \Phi_{\mathrm{Int}}(\delta))$-local correlated equilibrium or an $(\varepsilon, \Phi_{\mathrm{Proj}}(\delta))$-local correlated equilibrium, even when $\varepsilon = \Theta(1)$ and $G, L = O(\mathrm{poly}(d))$. Moreover, given black-box access to value and gradient queries, finding an $(\varepsilon, \Phi_{\mathrm{Int}}(\delta))$-local correlated equilibrium or an $(\varepsilon, \Phi_{\mathrm{Proj}}(\delta))$-local correlated equilibrium requires exponentially many queries in at least one of the parameters $d, G, L, 1/\varepsilon$. These results are presented as Theorem 6 and Theorem 7 in Appendix D.

We discuss additional related works in Appendix A.

## 2 PRELIMINARIES

A ball of radius $r > 0$ centered at $x \in \mathbb{R}^d$ is denoted by $B_d(x, r) := \{x' \in \mathbb{R}^d : \|x - x'\| \leq r\}$. We use $\|\cdot\|$ for $L_2$ norm throughout. We also write $B_d(\delta)$ for a ball centered at the origin with radius $\delta$. For $a \in \mathbb{R}$, we use $[a]^+$ to denote $\max\{0, a\}$. We denote $D_{\mathcal{X}}$ the diameter of a set $\mathcal{X}$.

**Differentiable / Smooth Games** An $n$-player *differentiable game* has a set of $n$ players $[n] := \{1, 2, \ldots, n\}$. Each player $i \in [n]$ has a nonempty convex and compact strategy set $\mathcal{X}_i \subseteq \mathbb{R}^{d_i}$. For a joint strategy profile $x = (x_i, x_{-i}) \in \prod_{j=1}^n \mathcal{X}_j$, the reward of player $i$ is determined by a utility function $u_i : \prod_{j=1}^n \mathcal{X}_j \to \mathbb{R}$ whose gradient with respect to $x_i$ is continuous. We denote $d = \sum_{i=1}^n d_i$ as the dimensionality of the game and assume $\max_{i \in [n]}\{D_{\mathcal{X}_i}\} \leq D$. A *smooth game* is a differentiable game whose utility functions further satisfy the following assumption.

**Assumption 1** (Smooth Games). *The utility function $u_i(x_i, x_{-i})$ for any player $i \in [n]$ satisfies:*

1. *(G-Lipschitzness) $\|\nabla_{x_i} u_i(x)\| \leq G$ for all $i$ and $x \in \prod_{j=1}^n \mathcal{X}_j$;*

2. *(L-smoothness) there exists $L_i > 0$ such that $\|\nabla_{x_i} u_i(x_i, x_{-i}) - \nabla_{x_i} u_i(x'_i, x_{-i})\| \leq L_i \|x_i - x'_i\|$ for all $x_i, x'_i \in \mathcal{X}_i$ and $x_{-i} \in \Pi_{j \neq i} \mathcal{X}_j$. We denote $L = \max_i L_i$ as the smoothness of the game.*

Crucially, we make no assumption on the concavity of $u_i(x_i, x_{-i})$.

**Local Nash Equilibrium** For $\varepsilon, \delta > 0$, an $(\varepsilon, \delta)$-local Nash equilibrium is a strategy profile in which no player can increase their own utility by more than $\varepsilon$ via a deviation bounded by $\delta$. The formal definition is as follows.

**Definition 1** ($(\varepsilon, \delta)$-Local Nash Equilibrium [DSZ21; Das22]). *In a smooth game, for some $\varepsilon, \delta > 0$, a strategy profile $x^* \in \prod_{j=1}^n \mathcal{X}_j$ is an $(\varepsilon, \delta)$-local Nash equilibrium if and only if for every player $i \in [n]$, $u_i(x_i, x^*_{-i}) \leq u_i(x^*) + \varepsilon$, for every $x_i \in B_{d_i}(x_i^*, \delta) \cap \mathcal{X}_i$; or equivalently, for every player $i \in [n]$, $\max_{v \in B_{d_i}(\delta)} u_i(\Pi_{\mathcal{X}_i}[x_i^* - v], x^*_{-i}) \leq u_i(x^*) + \varepsilon$.*

For large enough $\delta$, Definition 1 captures $\epsilon$-global Nash equilibrium as well. The notion of $(\varepsilon, \delta)$-local Nash equilibrium transitions from being $\varepsilon$-approximate local Nash equilibrium to $\varepsilon$-approximate Nash equilibrium as $\delta$ ranges from small to large. The complexity of computing of an $(\varepsilon, \delta)$-local Nash equilibrium is characterized by $\delta$ as follows (see Figure 1 for a summary).

- **Trivial Regime.** When $\delta < \varepsilon/G$, then every point $x \in \prod_{j=1}^n \mathcal{X}_j$ is an $(\varepsilon, \delta)$-local Nash equilibrium since for any player $i \in [n]$, it holds that $\|u_i(x) - u_i(x_i', x_{-i})\| \leq \varepsilon$ for every $x_i \in B_{d_i}(x_i, \delta)$.

- **Local Regime.** When $\delta \leq \sqrt{2\varepsilon/L}$, an $(\varepsilon, \delta)$-local Nash equilibrium always exists. However, finding an $(\varepsilon, \delta)$-local Nash equilibrium is PPAD-hard for any $\delta \geq \sqrt{\varepsilon/L}$, even when $1/\varepsilon, G, L = O(\text{poly}(d))$ [DSZ21]. Our main focus in this paper is the local regime.

- **Global Regime.** When $\delta \geq D$, then $(\varepsilon, \delta)$-local Nash equilibrium becomes a standard $\varepsilon$-Nash equilibrium and is NP-hard to find even if $\varepsilon = \Theta(1)$ and $G, L = O(\text{poly}(d))$ [DSZ21].

## 3 LOCAL CORRELATED EQUILIBRIUM AND $\Phi$-REGRET

In this section, we introduce the concept of local correlated equilibrium and explore its relationship with online learning and $\Phi$-regret. We provide two instantiations of our local correlated equilibrium and show that both instantiations are computationally tractable.

### 3.1 LOCAL CORRELATED EQUILIBRIUM

For $\delta > 0$, denote $\Phi(\delta) = \Pi_{i=1}^n \Phi^{\mathcal{X}_i}(\delta)$ be a collection of *local* strategy modifications such that for each $i \in [n]$, $\|\phi_i(x) - x\| \leq \delta$ for all $x \in \mathcal{X}_i$ and $\phi_i \in \Phi^{\mathcal{X}_i}(\delta)$. We define $(\varepsilon, \Phi(\delta))$-local correlated equilibrium of a differentiable game as a distribution over joint strategy profiles such that no player $i$ can increase their expected utility by more than $\varepsilon$ using any strategy modification in $\Phi^{\mathcal{X}_i}(\delta)$.

**Definition 2** (Local Correlated Equilibrium). *In a differentiable game, a distribution $\sigma$ over joint strategy profiles $\mathcal{X}$ is an $(\varepsilon, \Phi(\delta))$-local correlated equilibrium for some $\varepsilon, \delta > 0$ if and only if for all player $i \in [n]$, $\max_{\phi_i \in \Phi^{\mathcal{X}_i}(\delta)} \mathbb{E}_{x \sim \sigma}[u_i(\phi_i(x_i), x_{-i})] \leq \mathbb{E}_{x \sim \sigma}[u_i(x)] + \varepsilon$.*

By the definition of local modification, any $(\varepsilon, \delta)$-local Nash equilibrium is also an $(\varepsilon, \Phi(\delta))$-local correlated equilibrium for any set $\Phi(\delta)$. Thus in the local regime where $\delta \leq \sqrt{2\varepsilon/L}$, the existence of $(\varepsilon, \Phi(\delta))$-local correlated equilibrium for any set of $\Phi(\delta)$ follows from the existence of $(\varepsilon, \delta)$-local Nash equilibrium, which is established in [DSZ21]. $(\varepsilon, \Phi(\delta))$-local correlated equilibrium is closely related to the notion of $\Phi$-regret minimization in online learning. We first present some background on online learning and $\Phi$-regret.

**Online Learning and $\Phi$-Regret** We consider the standard online learning setting: at each day $t \in [T]$, the learner chooses an action $x^t$ from a nonempty convex compact set $\mathcal{X} \subseteq \mathbb{R}^m$ and the adversary chooses a possibly non-convex loss function $f^t : \mathcal{X} \to \mathbb{R}$, then the learner suffers a loss $f^t(x^t)$ and receives gradient feedback $\nabla f^t(x^t)$. We make the same assumptions on $\{f^t\}_{t \in [T]}$ as in Assumption 1 that each $f^t$ is $G$-Lipschitz, and $L$-smooth. The classic goal of an online learning algorithm is to minimize the *external regret* defined as $\text{Reg}^T := \max_{x \in \mathcal{X}} \sum_{t=1}^T (f^t(x^t) - f^t(x))$. An algorithm is called *no-regret* if its external regret is sublinear in $T$. The notion of $\Phi$-regret generalizes external regret by allowing general strategy modifications.

**Definition 3** ($\Phi$-regret). *Let $\Phi$ be a set of strategy modification functions $\{\phi : \mathcal{X} \to \mathcal{X}\}$. For $T \geq 1$, the $\Phi$-regret of an online learning algorithm is $\text{Reg}_\Phi^T := \max_{\phi \in \Phi} \sum_{t=1}^T (f^t(x^t) - f^t(\phi(x^t)))$. We call an algorithm no $\Phi$-regret if its $\Phi$-regret is sublinear in $T$.*

Many classic notions of regret can be interpreted as $\Phi$-regret. For example, the external regret is $\Phi_{\text{ext}}$-regret where $\Phi_{\text{ext}}$ contains all constant strategy modifications $\phi_{x^*}(x) = x^*$. The *swap regret* on simplex $\Delta^m$ is $\Phi_{\text{swap}}$-regret where $\Phi_{\text{swap}}$ contains all linear transformations $\phi : \Delta^m \to \Delta^m$.

The main result of the section is a reduction from $(\varepsilon, \Phi(\delta))$-local correlated equilibrium computation for *non-concave* smooth games in the local regime, to $\Phi^{\mathcal{X}_i}(\delta)$-regret minimization against *convex* losses. The key observation here is that the $L$-smoothness of the utility function permits the approximation of a non-concave function with a linear function within a local area bounded by $\delta$. This approximation yields an error of at most $\frac{\delta^2 L}{2}$, which is less than $\varepsilon$. We defer the proof to Appendix B.

**Lemma 1** (No $\Phi^{\mathcal{X}}(\delta)$-Regret to $(\varepsilon, \Phi(\delta))$-Local Correlated Equilibrium). *For any $T \geq 1$ and $\delta > 0$, let $\mathcal{A}$ be an online algorithm with $\Phi^{\mathcal{X}}(\delta)$-regret guarantee $\text{Reg}^T_{\Phi^{\mathcal{X}}(\delta)}$ for* convex *loss functions. Then*

1. *The $\Phi^{\mathcal{X}}(\delta)$-regret of $\mathcal{A}$ for* non-convex *and $L$-smooth loss functions is at most $\text{Reg}^T_{\Phi^{\mathcal{X}}(\delta)} + \frac{\delta^2 L T}{2}$.*

2. *When every player employs $\mathcal{A}$ in a non-concave $L$-smooth game, their empirical distribution of the joint strategies played converges to a $(\max_{i \in [n]}\{\text{Reg}^T_{\Phi^{\mathcal{X}_i}(\delta)}\} \cdot T^{-1} + \frac{\delta^2 L}{2}, \Phi(\delta))$-local correlated equilibrium.*

**Naive $\Phi$-Regret Minimization** By Lemma 1, it suffices to design no $\Phi$-regret algorithms against convex losses for efficient equilibrium computation. Although $\Phi$-regret minimization is extensively studied [GJ03; SL07; GGM08], to our knowledge, there is no efficient approach for a general set $\Phi(\delta) \subseteq \{\phi : \mathcal{X} \to \mathcal{X} \text{ and } \|\phi(x) - x\| \leq \delta\}$ of strategy modifications. By assuming Lipschitzness and the existence of fixed points for all $\phi \in \Phi$, a generic way for $\Phi$-regret minimization [GGM08] is: (1) discretize $\Phi$ and get a finite set $\Phi^\gamma$ with discretization error $\gamma > 0$; (2) apply an expert algorithm [FS97] over $\Phi^\gamma$. For $G$-Lipschitz and convex loss functions, this approach leads to $O(\delta G \sqrt{T \log |\Phi^\gamma|} + \gamma G T)$ $\Phi$-regret, since the loss range of the expert problem (i.e., the difference between the maximum and the minimum loss of the experts in each round) is at most $G\delta$. However, $|\Phi^\gamma|$ can be doubly exponential in the dimension of $\mathcal{X}$ and the per-iteration computational complexity (for running the expert algorithm and computing a fixed point) is prohibitively high.

### 3.2 Two Instantiations of the Local Correlated Equilibrium

In this section, we introduce two natural sets of local strategy modifications and show how to efficiently minimize the corresponding $\Phi(\delta)$-regret.

The first set $\Phi^{\mathcal{X}}_{\text{Int}}(\delta)$ is defined as follows: for $\delta \leq D_{\mathcal{X}}$ and $\lambda \in [0, 1]$, each strategy modification $\phi_{\lambda, x^*}$ interpolates the input strategy $x$ with a fixed strategy $x^*$: $\Phi^{\mathcal{X}}_{\text{Int}}(\delta) := \{\phi_{\lambda, x^*}(x) = (1 - \lambda)x + \lambda x^* : x^* \in \mathcal{X}, \lambda \leq \delta/D_{\mathcal{X}}\}$. Note that for any $x^* \in \mathcal{X}$ and $\lambda \leq \frac{\delta}{D_{\mathcal{X}}}$, we have $\|\phi_{\lambda, x^*}(x) - x\| = \lambda \|x - x^*\| \leq \delta$, respecting the locality requirement. The induced $\Phi^{\mathcal{X}}_{\text{Int}}(\delta)$-regret can be equivalently defined as $\text{Reg}^T_{\text{Int}, \delta} := \max_{x^* \in \mathcal{X}, \lambda \leq \frac{\delta}{D_{\mathcal{X}}}} \sum_{t=1}^T (f^t(x^t) - f^t((1 - \lambda)x^t + \lambda x^*))$.

The second set $\Phi^{\mathcal{X}}_{\text{Proj}}(\delta)$ encompasses all deviations that essentially add a fixed displacement vector $v$ to the input strategy: $\Phi^{\mathcal{X}}_{\text{Proj}}(\delta) := \{\phi_v(x) = \Pi_{\mathcal{X}}[x - v] : v \in B_d(\delta)\}$. It is clear that $\|\phi_v(x) - x\| \leq \|v\| \leq \delta$. The induced $\Phi^{\mathcal{X}}_{\text{Proj}}(\delta)$-regret is $\text{Reg}^T_{\text{Proj}, \delta} := \max_{v \in B_d(\delta)} \sum_{t=1}^T (f^t(x^t) - f^t(\Pi_{\mathcal{X}}[x^t - v]))$. The two sets of local strategy modifications above naturally induce two notions of local correlated equilibrium.

**Definition 4** (Two Instantiations of Local Correlated Equilibrium). *Let $\varepsilon, \delta > 0$. Define $\Phi_{\text{Int}}(\delta) = \Pi^n_{j=1} \Phi^{\mathcal{X}_j}_{\text{Int}}(\delta)$ and $\Phi_{\text{Proj}}(\delta) = \Pi^n_{j=1} \Phi^{\mathcal{X}_j}_{\text{Proj}}(\delta)$. In a smooth game, a distribution $\sigma$ over joint strategy profiles is an $(\varepsilon, \Phi_{\text{Int}}(\delta))$-local correlated equilibrium if and only if for all player $i \in [n]$,*

$$\max_{x' \in \mathcal{X}_i, \lambda \leq \delta/D_{\mathcal{X}_i}} \mathbb{E}_{x \sim \sigma}[u_i((1 - \lambda)x_i + \lambda x', x_{-i})] \leq \mathbb{E}_{x \sim \sigma}[u_i(x)] + \varepsilon.$$

*Similarly, $\sigma$ is an $(\varepsilon, \Phi_{\text{Proj}}(\delta))$-local correlated equilibrium if and only if for all player $i \in [n]$,*

$$\max_{v \in B_{d_i}(\delta)} \mathbb{E}_{x \sim \sigma}[u_i(\Pi_{\mathcal{X}_i}[x_i - v], x_{-i})] \leq \mathbb{E}_{x \sim \sigma}[u_i(x)] + \varepsilon.$$

Intuitively speaking, when a correlation device recommends a strategy to each player according to an $(\varepsilon, \Phi_{\text{Int}}(\delta))$-local correlated equilibrium, no player can increase their utility by more than $\varepsilon$ through a local deviation by interpolating with a fixed strategy. In contrast, an $(\varepsilon, \Phi_{\text{Proj}}(\delta))$-local correlated equilibrium guarantees no player can increase their utility by more than $\varepsilon$ through a fixed-direction local deviation.

One might be tempted to apply the previously mentioned discretization approach for $\Phi_{\text{Int}}^{\mathcal{X}}(\delta)$-regret or $\Phi_{\text{Proj}}^{\mathcal{X}}(\delta)$-regret minimization. However, for $\Phi_{\text{Int}}^{\mathcal{X}}(\delta)$ and $\Phi_{\text{Proj}}^{\mathcal{X}}(\delta)$, the discretized set with error $\gamma$ has size $O((\frac{D_{\mathcal{X}}}{\gamma})^m)$ or $O((\frac{\delta}{\gamma})^m)$ that is exponential in the dimension $m$. This leads to a regret of $\tilde{O}(\delta G \sqrt{mT})$ after choosing the optimal $\gamma$, but the per-iteration computational complexity is exponential in the dimension. In Section 4 and Section 5, we show how to minimize $\Phi$-regret for both $\Phi_{\text{Int}}^{\mathcal{X}}(\delta)$ and $\Phi_{\text{Proj}}^{\mathcal{X}}(\delta)$ using simple algorithms that are computationally efficient.

# 4    $\Phi_{\text{Int}}^{\mathcal{X}}(\delta)$-REGRET MINIMIZATION

In this section, we show how to achieve an $O(\sqrt{T})$ $\Phi_{\text{Int}}^{\mathcal{X}}(\delta)$-regret when the loss functions are convex. Our algorithms are simple and computationally efficient. Due to Lemma 1, these algorithms can be used to compute an $(\varepsilon, \Phi_{\text{Int}}(\delta))$-local correlated equilibrium.

Using the convexity of the loss functions $\{f^t\}_{t \in [T]}$, we have

$$\max_{\phi \in \Phi_{\text{Int}}^{\mathcal{X}}(\delta)} \sum_{t=1}^{T} \left( f^t(x^t) - f^t(\phi(x^t)) \right) = \max_{x^* \in \mathcal{X}, \lambda \leq \frac{\delta}{D_{\mathcal{X}}}} \sum_{t=1}^{T} \left( f^t(x^t) - f^t((1-\lambda)x^t + \lambda x^*) \right)$$

$$\leq \frac{\delta}{D_{\mathcal{X}}} \left[ \max_{x^* \in \mathcal{X}} \sum_{t=1}^{T} \left\langle \nabla f^t(x^t), x^t - x^* \right\rangle \right]^{+}.$$

Therefore, minimizing $\Phi_{\text{Int}}^{\mathcal{X}}(\delta)$-regret against convex loss functions can be reduced to minimizing the external regret with respect to linear loss functions. Note that when all $f^t$'s are linear functions, the reduction is without loss. Thus the worst-case $\Omega(D_{\mathcal{X}} G \sqrt{T})$ lower bound for external regret implies a $\Omega(\delta G \sqrt{T})$ lower bound for $\Phi_{\text{Int}}^{\mathcal{X}}(\delta)$-regret.

**Theorem 1.** *Let $\mathcal{A}$ be a algorithm with external regret $\text{Reg}^T(G, D_{\mathcal{X}})$ for linear and $G$-Lipschitz loss functions over $\mathcal{X}$. Then, for any $\delta > 0$, the $\Phi_{\text{Int}}^{\mathcal{X}}(\delta)$-regret of $\mathcal{A}$ for convex and $G$-Lipschitz loss functions over $\mathcal{X}$ is at most $\frac{\delta}{D_{\mathcal{X}}} \cdot [\text{Reg}^T(G, D_{\mathcal{X}})]^{+}$. Consequently, for the Online Gradient Descent algorithm (GD) [Zin03] with step size $\eta = \frac{D_{\mathcal{X}}}{G \sqrt{T}}$, its $\Phi_{\text{Int}}^{\mathcal{X}}(\delta)$-regret is at most $2\delta G \sqrt{T}$. Furthermore, for any $\delta > 0$ and any $\varepsilon > \frac{\delta^2 L}{2}$, when all players employ the GD algorithm in a smooth game, their empirical distribution of played strategy profiles converges to $(\varepsilon, \Phi_{\text{Int}}(\delta))$-local correlated equilibrium in $\frac{16\delta^2 G^2}{(2\varepsilon - \delta^2 L)^2}$ iterations.*

The above $O(\sqrt{T})$ $\Phi_{\text{Int}}^{\mathcal{X}}(\delta)$-regret bound is derived for the adversarial setting. In the game setting, where each player employs the same algorithm, players may have substantially lower external regret [Syr+15; CP20; DFG21; Ana+22a; Ana+22b; Far+22] but we need a slightly stronger smoothness assumption than Assumption 1. This assumption is naturally satisfied by normal-form games and is also made for results about concave games [Far+22].

**Assumption 2.** *For any player $i \in [n]$, the utility $u_i(x)$ satisfies $\|\nabla_{x_i} u_i(x) - \nabla_{x_i} u_i(x')\| \leq L \|x - x'\|$ for all $x, x' \in \mathcal{X}$.*

Using Assumption 2 and Lemma 1, the no-regret learning dynamics of [Far+22] that guarantees $O(\log T)$ individual external regret in concave games can be applied to smooth non-concave games so that the individual $\Phi_{\text{Int}}^{\mathcal{X}}(\delta)$-regret of each player is at most $O(\log T) + \frac{\delta^2 L T}{2}$. It then gives an algorithm for computing $(\varepsilon, \Phi_{\text{Int}}(\delta))$-local correlated equilibrium with faster convergence than GD.

# 5 $\Phi_{\mathrm{Proj}}^{\mathcal{X}}(\delta)$-REGRET MINIMIZATION

In this section, we investigate the $\Phi_{\mathrm{Proj}}^{\mathcal{X}}(\delta)$-regret. Unlike the $\Phi_{\mathrm{Int}}^{\mathcal{X}}(\delta)$-regret, we can not directly reduce $\Phi_{\mathrm{Proj}}^{\mathcal{X}}(\delta)$-regret minimization to external regret minimization. Recall the definition of $\Phi_{\mathrm{Proj}}^{\mathcal{X}}(\delta)$-regret that compares the cumulative loss and the loss of a fixed-direction local deviation: $\mathrm{Reg}_{\mathrm{Proj},\delta}^T := \max_{v \in B_d(\delta)} \sum_{t=1}^T \left( f^t(x^t) - f^t(\Pi_{\mathcal{X}}[x^t - v]) \right)$. Below, we first illustrate that external regret and $\Phi_{\mathrm{Proj}}^{\mathcal{X}}(\delta)$-regret are incomparable, and why a reduction similar to the one we show in Section 4 is unlikely. Next, we demonstrate, quite surprisingly, that classical algorithms like Online Gradient Descent (GD) and Optimistic Gradient (OG), known for minimizing external regret, also attain near-optimal $\Phi_{\mathrm{Proj}}^{\mathcal{X}}(\delta)$-regret. Missing proofs of this section are in Appendix C.

**Difference between external regret and $\Phi_{\mathrm{Proj}}^{\mathcal{X}}(\delta)$-regret** In the following two examples, we show that $\Phi_{\mathrm{Proj}}^{\mathcal{X}}(\delta)$-regret is incomparable with external regret for convex loss functions . A sequence of actions may suffer high $\mathrm{Reg}^T$ but low $\mathrm{Reg}_{\mathrm{Proj},\delta}^T$ (Example 1), and vise versa (Example 2).

**Example 1.** *Let $f^1(x) = f^2(x) = |x|$ for $x \in \mathcal{X} = [-1, 1]$. Then the $\Phi_{\mathrm{Proj}}^{\mathcal{X}}(\delta)$-regret of the sequence $\{x^1 = \frac{1}{2}, x^2 = -\frac{1}{2}\}$ for any $\delta \in (0, \frac{1}{2})$ is 0. However, the external regret of the same sequence is 1. By repeating the construction for $\frac{T}{2}$ times, we conclude that there exists a sequence of actions with $\mathrm{Reg}_{\mathrm{Proj},\delta}^T = 0$ and $\mathrm{Reg}^T = \frac{T}{2}$ for all $T \geq 2$.*

**Example 2.** *Let $f^1(x) = -2x$ and $f^2(x) = x$ for $x \in \mathcal{X} = [-1, 1]$. Then the $\Phi_{\mathrm{Proj}}^{\mathcal{X}}(\delta)$-regret of the sequence $\{x^1 = \frac{1}{2}, x^2 = 0\}$ for any $\delta \in (0, \frac{1}{2})$ is $\delta$. However, the external regret of the same sequence is 0. By repeating the construction for $\frac{T}{2}$ times, we conclude that there exists a sequence of actions with $\mathrm{Reg}_{\mathrm{Proj},\delta}^T = \frac{\delta T}{2}$ and $\mathrm{Reg}^T = 0$ for all $T \geq 2$.*

At a high level, the external regret competes against a fixed action, whereas $\Phi_{\mathrm{Proj}}^{\mathcal{X}}(\delta)$-regret is more akin to the notion of *dynamic regret*, competing with a sequence of varying actions. When the environment is stationary, i.e., $f^t = f$ (Example 1), a sequence of actions that are far away from the global minimum must suffer high regret, but may produce low $\Phi_{\mathrm{Proj}}^{\mathcal{X}}(\delta)$-regret since the change to the cumulative loss caused by a fixed-direction deviation could be neutralized across different actions in the sequence. In contrast, in a non-stationary (dynamic) environment (Example 2), every fixed action performs poorly, and a sequence of actions could suffer low regret against a fixed action but the $\Phi_{\mathrm{Proj}}^{\mathcal{X}}(\delta)$-regret that competes with a fixed-direction deviation could be large. The fact that small external regret does not necessarily equate to small $\Phi_{\mathrm{Proj}}^{\mathcal{X}}(\delta)$-regret is in sharp contrast to the behavior of the $\Phi_{\mathrm{Int}}^{\mathcal{X}}(\delta)$-regret. Nevertheless, despite these differences between the two notions of regret as shown above, they are *compatible* for convex loss functions: our main results in this section provide algorithms that minimize external regret and $\Phi_{\mathrm{Proj}}^{\mathcal{X}}(\delta)$-regret simultaneously.

## 5.1 $\Phi_{\mathrm{Proj}}^{\mathcal{X}}(\delta)$-REGRET MINIMIZATION IN THE ADVERSARIAL SETTING

In this section, we show that the classic Online Gradient Descent (GD) algorithm enjoys an $O(G\sqrt{\delta D_{\mathcal{X}} T})$ $\Phi_{\mathrm{Proj}}^{\mathcal{X}}(\delta)$-regret despite the difference between the external regret and $\Phi_{\mathrm{Proj}}^{\mathcal{X}}(\delta)$-regret. First, let us recall the update rule of GD: given initial point $x^1 \in \mathcal{X}$ and step size $\eta > 0$, GD updates in each iteration $t$:

$$x^{t+1} = \Pi_{\mathcal{X}}[x - \eta \nabla f^t(x^t)]. \tag{GD}$$

The key step in our analysis for GD is simple but novel and general. We can extend the analysis to many other algorithms such as Optimistic Gradient (OG) in Section 5.2.

**Theorem 2.** *Let $\delta > 0$ and $T \in \mathbb{N}$. For convex and $G$-Lipschitz loss functions $\{f^t : \mathcal{X} \to \mathbb{R}\}_{t \in [T]}$, the $\Phi_{\mathrm{Proj}}^{\mathcal{X}}(\delta)$-regret of (GD) with step size $\eta > 0$ is $\mathrm{Reg}_{\mathrm{Proj},\delta}^T \leq \frac{\delta^2}{2\eta} + \frac{\eta}{2} G^2 T + \frac{\delta D_{\mathcal{X}}}{\eta}$. We can choose $\eta$ optimally as $\frac{\sqrt{\delta(\delta + D_{\mathcal{X}})}}{G\sqrt{T}}$ and attain $\mathrm{Reg}_{\mathrm{Proj},\delta}^T \leq 2G\sqrt{\delta(\delta + D_{\mathcal{X}})T}$. For any $\delta > 0$ and any $\varepsilon > \frac{\delta^2 L}{2}$, when all player employ GD in a $L$-smooth game, their empirical distribution of player strategy profiles converges to $(\varepsilon, \Phi_{\mathrm{Proj}}(\delta))$-local correlated equilibrium in $O(\frac{\delta D G^2}{(2\varepsilon - \delta^2 L)^2})$ iterations.*

**Remark 1.** *Note that* $\Phi_{\mathrm{Proj}}^{\mathcal{X}}(\delta)$*-regret can also be viewed as the* dynamic regret *[Zin03] with changing comparators* $\{p^t := \Pi_{\mathcal{X}}[x - v]\}$*. However, we remark that our analysis does not follow from standard* $O(\frac{(1+P_T)}{\eta} + \eta T)$ *dynamic regret bound of GD [Zin03] since* $P_T$*, defined as* $\sum_{t=2}^{T} \|p^t - p^{t-1}\|$*, can be* $\Omega(\eta T)$*.*

*Proof.* Let us denote $v \in B_d(\delta)$ a fixed deviation and define $p^t = \Pi_{\mathcal{X}}[x^t - v]$. By standard analysis of GD [Zin03] (see also the proof of [Bub+15, Theorem 3.2] ), we have

$$\sum_{t=1}^{T} \left(f^t(x^t) - f^t(p^t)\right) \leq \sum_{t=1}^{T} \frac{1}{2\eta}\left(\left\|x^t - p^t\right\|^2 - \left\|x^{t+1} - p^t\right\|^2 + \eta^2\left\|\nabla f^t(x^t)\right\|^2\right)$$

$$\leq \sum_{t=1}^{T-1} \frac{1}{2\eta}\left(\left\|x^{t+1} - p^{t+1}\right\|^2 - \left\|x^{t+1} - p^t\right\|^2\right) + \frac{\delta^2}{2\eta} + \frac{\eta}{2}G^2 T,$$

where the last step uses $\|x^1 - p^1\| \leq \delta$ and $\|\nabla f^t(x^t)\| \leq G$. Here the terms $\|x^{t+1} - p^{t+1}\|^2 - \|x^{t+1} - p^t\|^2$ do not telescope, and we further relax them in the following key step.

**Key Step:** We relax the first term as:

$$\left\|x^{t+1} - p^{t+1}\right\|^2 - \left\|x^{t+1} - p^t\right\|^2 = \left\langle p^t - p^{t+1}, 2x^{t+1} - p^t - p^{t+1}\right\rangle$$

$$= \left\langle p^t - p^{t+1}, 2x^{t+1} - 2p^{t+1}\right\rangle - \left\|p^t - p^{t+1}\right\|^2$$

$$= 2\left\langle p^t - p^{t+1}, v\right\rangle + 2\left\langle p^t - p^{t+1}, x^{t+1} - v - p^{t+1}\right\rangle - \left\|p^t - p^{t+1}\right\|^2$$

$$\leq 2\left\langle p^t - p^{t+1}, v\right\rangle - \left\|p^t - p^{t+1}\right\|^2,$$

where in the last inequality we use the fact that $p^{t+1}$ is the projection of $x^{t+1} - v$ onto $\mathcal{X}$ and $p^t$ is in $\mathcal{X}$. Now we get a telescoping term $2\langle p^t - p^{t+1}, u\rangle$ and a negative term $-\|p^t - p^{t+1}\|^2$. The negative term is useful for improving the regret analysis in the game setting, but we ignore it for now. Combining the two inequalities above, we have

$$\sum_{t=1}^{T}\left(f^t(x^t) - f^t(p^t)\right) \leq \frac{\delta^2}{2\eta} + \frac{\eta}{2}G^2 T + \frac{1}{\eta}\sum_{t=1}^{T-1}\left\langle p^t - p^{t+1}, v\right\rangle$$

$$= \frac{\delta^2}{2\eta} + \frac{\eta}{2}G^2 T + \frac{1}{\eta}\left\langle p^1 - p^T, v\right\rangle \leq \frac{\delta^2}{2\eta} + \frac{\eta}{2}G^2 T + \frac{\delta D_{\mathcal{X}}}{\eta}.$$

Since the above holds for any $v$ with $\|v\| \leq \delta$, it also upper bounds $\mathrm{Reg}_{\mathrm{Proj},\delta}^T$. $\qquad\square$

**Lower bounds for** $\Phi_{\mathrm{Proj}}^{\mathcal{X}}(\delta)$**-regret** We complement our upper bound with two lower bounds for $\Phi_{\mathrm{Proj}}^{\mathcal{X}}(\delta)$-regret minimization. The first one is an $\Omega(\delta G\sqrt{T})$ lower bound for any online learning algorithms against linear loss functions. The proof of Theorem 3 is postpone to Appendix C.

**Theorem 3** (Lower bound for $\Phi_{\mathrm{Proj}}^{\mathcal{X}}(\delta)$-regret against convex losses)**.** *For any* $T \geq 1$*,* $D_{\mathcal{X}} > 0$*,* $0 < \delta \leq D_{\mathcal{X}}$*, and* $G \geq 0$*, there exists a distribution* $\mathcal{D}$ *on* $G$*-Lipschitz linear loss functions* $f^1, \ldots, f^T$ *over* $\mathcal{X} = [-D_{\mathcal{X}}, D_{\mathcal{X}}]$ *such that for any online algorithm, its* $\Phi_{\mathrm{Proj}}^{\mathcal{X}}(\delta)$*-regret on the loss sequence satisfies* $\mathbb{E}_{\mathcal{D}}[\mathrm{Reg}_{\mathrm{Proj},\delta}^T] = \Omega(\delta G\sqrt{T})$*. Note that linear functions are* $0$*-smooth, so the same lower bound holds for* $G$*-Lipschitz and* $L$*-smooth convex loss functions.*

**Remark 2.** *A keen reader may notice that the* $\Omega(G\delta\sqrt{T})$ *lower bound in Theorem 3 does not match the* $O(G\sqrt{\delta D_{\mathcal{X}} T})$ *upper bound in Theorem 2, especially when* $D_{\mathcal{X}} \gg \delta$*. A natural question is: which of them is tight? We conjecture that the lower bound is tight. In fact, for the special case where the feasible set* $\mathcal{X}$ *is a box, we obtain a* $D_{\mathcal{X}}$*-independent bound* $O(d^{\frac{1}{4}} G\delta\sqrt{T})$ *using a modified version of GD, which is tight when* $d = 1$*. See Appendix E for a detailed discussion.*

This lower bound suggests that GD achieves near-optimal $\Phi_{\mathrm{Proj}}^{\mathcal{X}}(\delta)$-regret for convex losses. For $L$-smooth non-convex loss functions, we provide another $\Omega(\delta^2 LT)$ lower bound for algorithms that

satisfy the linear span assumption. The *linear span* assumption states that the algorithm produces $x^{t+1} \in \{\Pi_{\mathcal{X}}[\sum_{i \in [t]} a_i \cdot x^i + b_i \cdot \nabla f^i(x^i)] : a_i, b_i \in \mathbb{R}, \forall i \in [t]\}$ as essentially the linear combination of the previous iterates and their gradients. Many online algorithms such as online gradient descent and optimistic gradient satisfy the linear span assumption. The intuition behind the $\Omega(\delta^2 LT)$ lower bound for $\Phi_{\mathrm{Proj}}^{\mathcal{X}}(\delta)$-regret lies in the behavior of any algorithm adhering to the linear span assumption. Such an algorithm, when initialized at a local maximum where the gradient is zero, could remain there. Nonetheless, a $\delta$-local deviation can reduce the loss by $\Omega(\delta^2 L)$. Combining with Lemma 1, this lower bound suggests that GD attains nearly optimal $\Phi_{\mathrm{Proj}}^{\mathcal{X}}(\delta)$-regret, even in the non-convex setting, among a natural family of gradient-based algorithms.

**Proposition 1** (Lower bound for $\Phi_{\mathrm{Proj}}^{\mathcal{X}}(\delta)$-regret against non-convex losses). *For any $T \geq 1$, $\delta \in (0, 1)$, and $L \geq 0$, there exists a sequence of $L$-Lipschitz and $L$-smooth non-convex loss functions $f^1, \ldots, f^T$ on $\mathcal{X} = [-1, 1]$ such that for any algorithm that satisfies the linear span assumption, its $\Phi_{\mathrm{Proj}}^{\mathcal{X}}(\delta)$-regret on the loss sequence is $\mathrm{Reg}_{\mathrm{Proj},\delta}^T \geq \frac{\delta^2 LT}{2}$.*

## 5.2 Improved $\Phi_{\mathrm{Proj}}^{\mathcal{X}}(\delta)$- Regret in the Game Setting

Any online algorithm, as demonstrated by Theorem 3, suffers an $\Omega(\sqrt{T})$ $\Phi_{\mathrm{Proj}}^{\mathcal{X}}(\delta)$-regret even against linear loss functions. This lower bound, however, holds only in the *adversarial* setting where an adversary can choose arbitrary loss functions. In this section, we show an improved $O(T^{\frac{1}{4}})$ individual $\Phi_{\mathrm{Proj}}^{\mathcal{X}}(\delta)$-regret bound under a stronger version of the smoothness assumption (Assumption 2) in the *game* setting, where players interact with each other using the same algorithm.

We study the Optimistic Gradient (OG) algorithm [RS13], an optimistic variant of GD that has been shown to have improved individual *external* regret guarantee in the game setting [Syr+15]. The OG algorithm initializes $w^0 \in \mathcal{X}$ arbitrarily and $g^0 = 0$. In each step $t \geq 1$, the algorithm plays $x^t$, receives feedback $g^t := \nabla f^t(x^t)$, and updates $w^t$, as follows:

$$x^t = \Pi_{\mathcal{X}}[w^{t-1} - \eta g^{t-1}], \quad w^t = \Pi_{\mathcal{X}}[w^{t-1} - \eta g^t]. \tag{OG}$$

We first prove an $O(\sqrt{T})$ $\Phi_{\mathrm{Proj}}^{\mathcal{X}}(\delta)$-regret upper bound for OG in the adversarial setting.

**Theorem 4** (Adversarial Regret Bound for OG). *Let $\delta > 0$ and $T \in \mathbb{N}$. For convex and $G$-Lipschitz loss functions $\{f^t : \mathcal{X} \to \mathbb{R}\}_{t \in [T]}$, the $\Phi_{\mathrm{Proj}}^{\mathcal{X}}(\delta)$-regret of (OG) with step size $\eta > 0$ is $\mathrm{Reg}_{\mathrm{Proj},\delta}^T \leq \frac{\delta D_{\mathcal{X}}}{\eta} + \eta \sum_{t=1}^T \|g^t - g^{t-1}\|^2$. Choosing step size $\eta = \frac{\sqrt{\delta D_{\mathcal{X}}}}{2G\sqrt{T}}$, we have $\mathrm{Reg}_{\mathrm{Proj},\delta}^T \leq 4G\sqrt{\delta D_{\mathcal{X}} T}$.*

In the analysis of Theorem 4 for the adversarial setting, the term $\|g^t - g^{t-1}\|^2$ can be as large as $4G^2$. In the game setting where every player $i$ employs OG, $g_i^t$, i.e., $-\nabla_{x_i} u_i(x)$, depends on other players' action $x_{-i}^t$. Note that the change of the players' actions $\|x^t - x^{t-1}\|^2$ is only $O(\eta^2)$. Such stability of the updates leads to an improved upper bound on $\|g_i^t - g_i^{t-1}\|^2$ and hence also an improved $O(T^{\frac{1}{4}})$ $\Phi_{\mathrm{Proj}}^{\mathcal{X}}(\delta)$-regret for the player.

**Theorem 5** (Improved Individual $\Phi_{\mathrm{Proj}}^{\mathcal{X}}(\delta)$-Regret of OG in the Game Setting). *In a $G$-Lipschitz $L$-smooth (in the sense of Assumption 2) differentiable game, when all players employ OG with step size $\eta > 0$, then for each player $i$, $\delta > 0$, and $T \geq 1$, their individual $\Phi_{\mathrm{Proj}}^{\mathcal{X}_i}(\delta)$-regret denoted as $\mathrm{Reg}_{\mathrm{Proj},\delta}^{T,i}$ is $\mathrm{Reg}_{\mathrm{Proj},\delta}^{T,i} \leq \frac{\delta D}{\eta} + \eta G^2 + 3nL^2 G^2 \eta^3 T$. Choosing $\eta = \min\{(\delta D/(nL^2 G^2 T))^{\frac{1}{4}}, (\delta D)^{\frac{1}{2}}/G\}$, we have $\mathrm{Reg}_{\mathrm{Proj},\delta}^{T,i} \leq 4(\delta D)^{\frac{3}{4}}(nL^2 G^2 T)^{\frac{1}{4}} + 2\sqrt{\delta D}G$. Furthermore, for any $\delta > 0$ and any $\varepsilon > \frac{\delta^2 L}{2}$, their empirical distribution of played strategy profiles converges to $(\varepsilon, \Phi_{\mathrm{Proj}}(\delta))$-local correlated equilibrium in $O(\max\{\frac{\delta D(nL^2 G^2)^{\frac{1}{3}}}{(2\varepsilon - \delta^2 L)^{\frac{4}{3}}}, \frac{\sqrt{\delta D}G}{2\varepsilon - \delta^2 L}\})$ iterations.*

## 6 Discussion and Future Directions

**More local $\Phi$-regret**  In this paper, we propose two natural set of $\Phi_{\mathrm{Int}}^{\mathcal{X}}(\delta)$ and $\Phi_{\mathrm{Proj}}^{\mathcal{X}}(\delta)$ with game-theoretical implications and efficient $\Phi$-regret minimization. It will be interesting to investigate which other local strategy modifications also have this property, i.e., the corresponding $\Phi$-regret can be minimized efficiently.

**Improved $\Phi_{\mathrm{Proj}}^{\mathcal{X}}(\delta)$-regret in games** We show in Theorem 5 that the optimistic gradient (OG) dynamics gives individual $\Phi_{\mathrm{Proj}}^{\mathcal{X}}(\delta)$-regret of $O(T^{1/4})$. Could we design uncoupled learning dynamics with better individual regret guarantee, consequently leading to faster convergence to an $(\varepsilon, \Phi_{\mathrm{Proj}}(\delta))$-local correlated equilibrium?

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

## Contents

## A  Additional Related Works

**Non-Concave Games.**    An important special case of multi-player games are two-player zero-sum games, which are defined in terms of some function $f : \mathcal{X} \times \mathcal{Y} \to \mathbb{R}$ that one of the two players, say

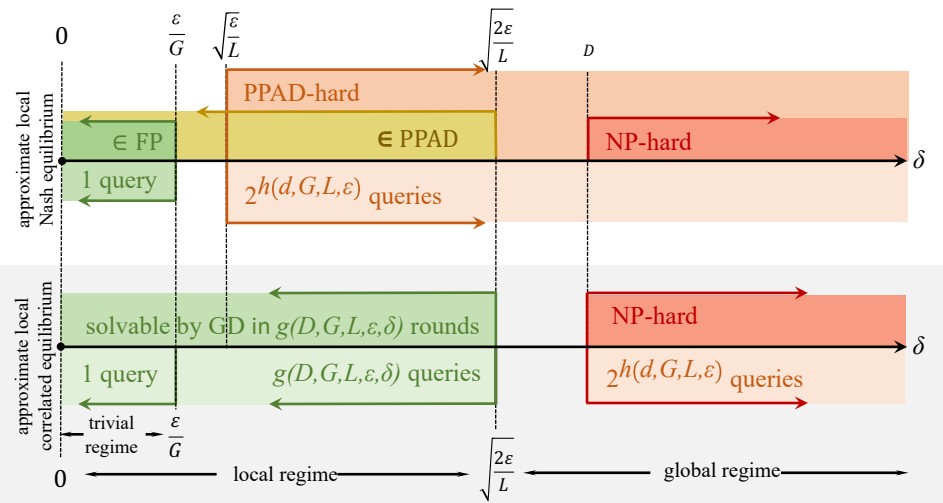

Figure 1: Summary of results and comparison between the complexity of computing an $(\varepsilon, \delta)$-local Nash equilibrium and an $(\varepsilon, \Phi(\delta))$-local correlated equilibrium (for $\Phi_{\mathrm{Int}}(\delta)$ and $\Phi_{\mathrm{Proj}}(\delta)$) in a $d$-dimensional, $G$-Lispchitz, and $L$-smooth game where each player's strategy space is bounded by $D$. We assume $\varepsilon \leq G^2/L$ so that the trivial regime is a subset of the local regime. The function $h$ in the query lower bound is defined as $h(d, G, L, \varepsilon) = (\min(d, G/\varepsilon, L/\varepsilon))^p$ for some universal constant $p > 0$. The function $g$ is defined as $g(D, G, L, \varepsilon, \delta) = \mathrm{poly}(D, G, L, \delta, \frac{1}{(2\varepsilon - \delta^2 L)})$. PPAD-hard problems are widely believed to be computationally intractable. Our main results are (1) efficient algorithms for computing both $(\varepsilon, \Phi_{\mathrm{Int}}(\delta))$ and $(\varepsilon, \Phi_{\mathrm{Proj}}(\delta))$-local correlated equilibrium in the local regime and (2) intractability for both $(\varepsilon, \Phi_{\mathrm{Int}}(\delta))$ and $(\varepsilon, \Phi_{\mathrm{Proj}}(\delta))$-local correlated equilibrium in the global regime.

the one choosing $x \in \mathcal{X}$, wants to minimize, while the other player, the one choosing $y \in \mathcal{Y}$, wants to maximize. Finding Nash equilibrium in such games is tractable in the *convex-concave* setting, i.e. when $f(x, y)$ is convex with respect to the minimizing player's strategy, $x$, and concave with respect to the maximizing player's strategy, $y$, but it is computationally intractable in the general *nonconvex-nonconcave* setting. Namely, a Nash equilibrium may not exist, and it is NP-hard to determine if one exists and, if so, find it. Moreover, in this case, stable limit points of gradient-based dynamics are not necessarily Nash equilibria, not even local Nash equilibria [DP18; MRS20]. There is a line of work focusing on computing Nash equilibrium under additional structure in the game. This encompasses settings where the game satisfies the (weak) Minty variational inequality [MZ19; DDJ21; Pet+22; CZ23], or is sufficiently close to being bilinear [ALW21]. However, the study of universal solution concepts in the nonconvex-nonconcave setting is sparse. Daskalakis, Skoulakis, and Zampetakis [DSZ21] proved the existence and computational hardness of local Nash equilibrium. In a more recent work, [Das+23] proposes second-order algorithms with asymptotic convergence to local Nash equilibrium. Several works study sequential two-player zero-sum games, and make additional assumptions about the player who goes second. They propose equilibrium concepts such as *local minimax points* [JNJ20], *differentiable Stackelberg equilibrium* [FCR20], and *greedy adversarial equilibrium* [MV21]. Notably, local minimax points are stable limit points of Gradient-Descent-Ascent (GDA) dynamics [JNJ20; WZB20; FR21] while greedy adversarial equilibrium can be computed efficiently using second-order algorithms in the unconstrained setting [MV21]. In contrast to these studies, we focus on the more general case of multi-player non-concave games.

**Online Learning with Non-Convex Losses** A line of work has studied online learning against non-convex losses. To circumvent the computational intractability of this problem, various approaches have been pursued: some works assume a restricted set of non-convex loss functions [GLZ18], while others assume access to a sampling oracle [MM10; Kri+15] or access to an offline optimization oracle [AGH19; SN20; Hél+20] or a weaker notion of regret [HSZ17; AZF19; HMC21; GZL23]. The work most closely related to ours is [HSZ17]. The authors propose a notion of *w-smoothed local regret* against non-convex losses, and they also define a local equilibrium

concept for non-concave games. They use the idea of *smoothing* to average the loss functions in the previous $w$ iterations and design algorithms with optimal $w$-smoothed local regret. The concept of regret they introduce suggests a local equilibrium concept. However, their local equilibrium concept is very non-standard in that its local stability is not with respect to a distribution over strategy profiles sampled by this equilibrium concept. Moreover, the path to attaining this local equilibrium through decentralized learning dynamics remains unclear. The algorithms provided in [HSZ17; GZL23] require that every agent $i$ experiences (over several rounds) the average utility function of the previous $w$ iterates, denoted as $F_{i,w}^t := \frac{1}{w} \sum_{\ell=0}^{w-1} u_i^{t-\ell}(\cdot, x_{-i}^{t-\ell})$. Implementing this imposes significant coordination burden on the agents. In contrast, we focus on a natural concept of local correlated equilibrium, which is incomparable to that of [HSZ17], and we also show that efficient convergence to this concept is achieved via decentralized gradient-based learning dynamics.

**$\Phi$-regret and $\Phi$-equilibrium** The concept of $\Phi$-regret and the associated $\Phi$-equilibrium is introduced by Greenwald and Jafari [GJ03] and has been broadly investigated in the context of concave games [GJ03; SL07; GGM08; Ber+23]. The two types of local correlated equilibrium we explore can be regarded as special cases of $\Phi$-equilibrium in non-concave games, wherein all strategy modifications are localized. To the best of our knowledge, no one has yet focused on this unique setting, and no efficient algorithm is known for minimizing such $\Phi$-regret or computing the corresponding $\Phi$-equilibrium.

## B  PROOF OF LEMMA 1

Let $\{f^t\}_{t \in [T]}$ be a sequence of non-convex $L$-smooth loss functions satisfying Assumption 1. Let $\{x^t\}_{t \in [T]}$ be the iterates produced by $\mathcal{A}$ against $\{f^t\}_{t \in [T]}$. Then $\{x^t\}_{t \in [T]}$ is also the iterates produced by $\mathcal{A}$ against a sequence of linear loss functions $\{\langle \nabla f^t(x^t), \cdot \rangle\}$. For the latter, we know

$$\max_{\phi \in \Phi^{\mathcal{X}}(\delta)} \sum_{t=1}^T \langle \nabla f^t(x^t), x^t - \phi(x^t) \rangle \leq \mathrm{Reg}_{\Phi^{\mathcal{X}}(\delta)}^T.$$

Then using $L$-smoothness of $\{f^t\}$ and the fact that $\|\phi(x) - x\| \leq \delta$ for all $\phi \in \Phi(\delta)$, we get

$$\max_{\phi \in \Phi^{\mathcal{X}}(\delta)} \sum_{t=1}^T f^t(x^t) - f^t(\phi(x^t)) \leq \max_{\phi \in \Phi^{\mathcal{X}}(\delta)} \sum_{t=1}^T \left( \langle \nabla f^t(x^t), x^t - \phi(x^t) \rangle + \frac{L}{2} \|x^t - \phi(x^t)\|^2 \right)$$

$$\leq \mathrm{Reg}_{\Phi^{\mathcal{X}}(\delta)}^T + \frac{\delta^2 L T}{2}.$$

This completes the proof of the first part.

Let each player $i \in [n]$ employ algorithm $\mathcal{A}$ in a smooth game independently and produces iterates $\{x^t\}$. The averaged joint strategy profile $\sigma^T$ that chooses $x^t$ uniformly at random from $t \in [T]$ satisfies for any player $i \in [n]$,

$$\max_{\phi \in \Phi^{\mathcal{X}_i}(\delta)} \mathbb{E}_{x \sim \sigma}[u_i(\phi(x_i), x_{-i})] - \mathbb{E}_{x \sim \sigma}[u_i(x)]$$

$$= \max_{\phi \in \Phi^{\mathcal{X}_i}(\delta)} \frac{1}{T} \sum_{t=1}^T \left( u_i(\phi(x_i^t), x_{-i}^t) - u_i(x^t) \right)$$

$$\leq \frac{\mathrm{Reg}_{\Phi^{\mathcal{X}_i}(\delta)}^T}{T} + \frac{\delta^2 L}{2}.$$

Thus $\sigma^T$ is a $(\max_{i \in [n]} \{\mathrm{Reg}_{\Phi^{\mathcal{X}_i}(\delta)}^T\} \cdot T^{-1} + \frac{\delta^2 L}{2}, \Phi(\delta))$-local correlated equilibrium. This completes the proof of the second part.

## C  MISSING PROOFS IN SECTION 5

### C.1  PROOF OF THEOREM 3

Our proof technique comes from the standard one used in multi-armed bandits [Aue+02, Theorem 5.1]. Suppose that $f^t(x) = g^t x$. We construct two possible environments. In the first environment,

$g^t = G$ with probability $\frac{1+\epsilon}{2}$ and $g^t = -G$ with probability $\frac{1-\epsilon}{2}$; in the second environment, $g^t = G$ with probability $\frac{1-\epsilon}{2}$ and $g^t = -G$ with probability $\frac{1+\epsilon}{2}$. We use $\mathbb{E}_i$ and $\mathbb{P}_i$ to denote the expectation and probability measure under environment $i$, respectively, for $i = 1, 2$. Suppose that the true environment is uniformly chosen from one of these two environments. Below, we show that the expected regret of the learner is at least $\Omega(\delta G\sqrt{T})$.

Define $N_+ = \sum_{t=1}^{T} \mathbb{I}\{x^t \geq 0\}$ be the number of times $x^t$ is non-negative, and define $f^{1:T} = (f^1, \ldots, f^T)$. Then we have

$$
|\mathbb{E}_1[N_+] - \mathbb{E}_2[N_+]| = \left| \sum_{f^{1:T}} \left( \mathbb{P}_1(f^{1:T})\mathbb{E}\left[N_+ \mid f^{1:T}\right] - \mathbb{P}_2(f^{1:T})\mathbb{E}\left[N_+ \mid f^{1:T}\right] \right) \right|
$$

$$
\text{(enumerate all possible sequences of } f^{1:T})
$$

$$
\leq T \sum_{f^{1:T}} \left| \mathbb{P}_1(f^{1:T}) - \mathbb{P}_2(f^{1:T}) \right|
$$

$$
= T \|\mathbb{P}_1 - \mathbb{P}_2\|_{\text{TV}}
$$

$$
\leq T\sqrt{(2\ln 2)\text{KL}(\mathbb{P}_1, \mathbb{P}_2)} \qquad \text{(Pinsker's inequality)}
$$

$$
= T\sqrt{(2\ln 2)T \cdot \text{KL}\left(\text{Bernoulli}\left(\frac{1+\epsilon}{2}\right), \text{Bernoulli}\left(\frac{1-\epsilon}{2}\right)\right)}
$$

$$
= T\sqrt{(2\ln 2)T\epsilon \ln \frac{1+\epsilon}{1-\epsilon}} \leq T\sqrt{(4\ln 2)T\epsilon^2}. \qquad (1)
$$

In the first environment, we consider the regret with respect to $v = \delta$. Then we have

$$
\mathbb{E}_1\left[\text{Reg}_{\text{Proj},\delta}^T\right] \geq \mathbb{E}_1\left[\sum_{t=1}^{T} f^t(x^t) - f^t(\Pi_{\mathcal{X}}[x^t - \delta])\right] = \mathbb{E}_1\left[\sum_{t=1}^{T} g^t(x^t - \Pi_{\mathcal{X}}[x^t - \delta])\right]
$$

$$
= \mathbb{E}_1\left[\sum_{t=1}^{T} \epsilon G(x^t - \Pi_{\mathcal{X}}[x^t - \delta])\right] \geq \epsilon\delta G\mathbb{E}_1\left[\sum_{t=1}^{T} \mathbb{I}\{x^t \geq 0\}\right] = \epsilon\delta G\mathbb{E}_1\left[N_+\right],
$$

where in the last inequality we use the fact that if $x^t \geq 0$ then $x^t - \Pi_{\mathcal{X}}[x^t - \delta] = x^t - (x^t - \delta) = \delta$ because $D \geq \delta$. In the second environment, we consider the regret with respect to $v = -\delta$. Then similarly, we have

$$
\mathbb{E}_2\left[\text{Reg}_{\text{Proj},\delta}^T\right] \geq \mathbb{E}_2\left[\sum_{t=1}^{T} f^t(x^t) - f^t(\Pi_{\mathcal{X}}[x^t + \delta])\right] = \mathbb{E}_2\left[\sum_{t=1}^{T} g^t(x^t - \Pi_{\mathcal{X}}[x^t + \delta])\right]
$$

$$
= \mathbb{E}_2\left[\sum_{t=1}^{T} -\epsilon G(x^t - \Pi_{\mathcal{X}}[x^t + \delta])\right] \geq \epsilon\delta G\mathbb{E}_2\left[\sum_{t=1}^{T} \mathbb{I}\{x^t < 0\}\right] = \epsilon\delta G\left(T - \mathbb{E}_2\left[N_+\right]\right).
$$

Summing up the two inequalities, we get

$$
\frac{1}{2}\left(\mathbb{E}_1\left[\text{Reg}_{\text{Proj},\delta}^T\right] + \mathbb{E}_2\left[\text{Reg}_{\text{Proj},\delta}^T\right]\right) \geq \frac{1}{2}\left(\epsilon\delta GT + \epsilon\delta G(\mathbb{E}_1[N_+] - \mathbb{E}_2[N_+])\right)
$$

$$
\geq \frac{1}{2}\left(\epsilon\delta GT - \epsilon\delta GT\epsilon\sqrt{(4\ln 2)T}\right). \qquad \text{(by (1))}
$$

Choosing $\epsilon = \frac{1}{\sqrt{(16\ln 2)T}}$, we can lower bound the last expression by $\Omega(\delta G\sqrt{T})$. The theorem is proven by noticing that $\frac{1}{2}\left(\mathbb{E}_1\left[\text{Reg}_{\text{Proj},\delta}^T\right] + \mathbb{E}_2\left[\text{Reg}_{\text{Proj},\delta}^T\right]\right)$ is the expected regret of the learner.

### C.2 Proof of Proposition 1

*Proof.* Consider $f : [-1, 1] \to \mathbb{R}$ such that $f(x) = -\frac{L}{2}x^2$ and let $f^t = f$ for all $t \in [T]$. Then any first-order methods that satisfy the linear span assumption with initial point $x^1 = 0$ will produce $x^t = 0$ for all $t \in [T]$. The $\Phi_{\text{Proj}}^{\mathcal{X}}(\delta)$-regret is thus $\sum_{t=1}^{T}(f(0) - f(\delta)) = \frac{\delta^2 LT}{2}$. $\qquad \square$

### C.3 Proof of Theorem 4

*Proof.* Fix any deviation $v$ that is bounded by $\delta$. Let us define $p^0 = w^0$ and $p^t = \Pi_{\mathcal{X}}[x^t - v]$. Following standard analysis of OG [RS13], we have

$$
\begin{aligned}
\sum_{t=1}^{T} f^t(x^t) - f^t(p^t) &\leq \sum_{t=1}^{T} \left\langle \nabla f^t(x^t), x^t - p^t \right\rangle \\
&\leq \sum_{t=1}^{T} \frac{1}{2\eta} \left( \left\| w^{t-1} - p^t \right\|^2 - \left\| w^t - p^t \right\|^2 \right) + \eta \left\| g^t - g^{t-1} \right\|^2 - \frac{1}{2\eta} \left( \left\| x^t - w^t \right\|^2 + \left\| x^t - w^{t-1} \right\|^2 \right) \\
&\leq \sum_{t=1}^{T} \left( \frac{1}{2\eta} \left\| w^{t-1} - p^t \right\|^2 - \frac{1}{2\eta} \left\| w^{t-1} - p^{t-1} \right\|^2 + \eta \left\| g^t - g^{t-1} \right\|^2 - \frac{1}{2\eta} \left\| x^t - w^{t-1} \right\|^2 \right) \quad (2)
\end{aligned}
$$

Now we apply similar analysis from Theorem 2 to upper bound the term $\left\| w^{t-1} - p^t \right\|^2 - \left\| w^{t-1} - p^{t-1} \right\|^2$:

$$
\begin{aligned}
\left\| w^{t-1} - p^t \right\|^2 &- \left\| w^{t-1} - p^{t-1} \right\|^2 \\
&= \left\langle p^{t-1} - p^t, 2w^{t-1} - p^{t-1} - p^t \right\rangle \\
&= \left\langle p^{t-1} - p^t, 2w^{t-1} - 2p^t \right\rangle - \left\| p^t - p^{t-1} \right\|^2 \\
&= 2\left\langle p^{t-1} - p^t, v \right\rangle + 2\left\langle p^{t-1} - p^t, w^{t-1} - v - p^t \right\rangle - \left\| p^t - p^{t-1} \right\|^2 \\
&= 2\left\langle p^{t-1} - p^t, v \right\rangle + 2\left\langle p^{t-1} - p^t, x^t - v - p^t \right\rangle + 2\left\langle p^{t-1} - p^t, w^{t-1} - x^t \right\rangle - \left\| p^t - p^{t-1} \right\|^2 \\
&\leq 2\left\langle p^{t-1} - p^t, v \right\rangle + \left\| x^t - w^{t-1} \right\|^2,
\end{aligned}
$$

where in the last-inequality we use $\left\langle p^{t-1} - p^t, x^t - u - p^t \right\rangle \leq 0$ since $p^t = \Pi_{\mathcal{X}}[x^t - v]$ and $\mathcal{X}$ is a compact convex set; we also use $2\langle a, b \rangle - b^2 \leq a^2$. In the analysis above, unlike the analysis of GD where we drop the negative term $-\left\| p^t - p^{t-1} \right\|^2$, we use $-\left\| p^t - p^{t-1} \right\|^2$ to get a term $\left\| x^t - w^{t-1} \right\|^2$ which can be canceled by the last term in (2).

Now we combine the above two inequalities. Since the term $\left\| x^t - w^{t-1} \right\|^2$ cancels out and $2\left\langle p^{t-1} - p^t, v \right\rangle$ telescopes, we get

$$
\sum_{t=1}^{T} f^t(x^t) - f^t(p^t) \leq \frac{\left\langle p^0 - p^T, u \right\rangle}{\eta} + \sum_{t=1}^{T} \eta \left\| g^t - g^{t-1} \right\|^2 \leq \frac{\delta D_{\mathcal{X}}}{\eta} + \eta \sum_{t=1}^{T} \left\| g^t - g^{t-1} \right\|^2 \quad \square
$$

### C.4 Proof of Theorem 5

*Proof.* Let us fix any player $i \in [n]$ in the smooth game. In every step $t$, player $i$'s loss function $f^t : \mathcal{X}_i \to \mathbb{R}$ is $\langle -\nabla_{x_i} u_i(x^t), \cdot \rangle$ determined by their utility function $u_i$ and all players' actions $x^t$. Therefore, their gradient feedback is $g^t = -\nabla_{x_i} u_i(x^t)$. For all $t \geq 2$, we have

$$
\begin{aligned}
\left\| g^t - g^{t-1} \right\|^2 &= \left\| \nabla u_i(x^t) - \nabla u_i(x^{t-1}) \right\|^2 \\
&\leq L^2 \left\| x^t - x^{t-1} \right\|^2 \\
&= L^2 \sum_{i=1}^{n} \left\| x_i^t - x_i^{t-1} \right\|^2 \\
&\leq 3L^2 \sum_{i=1}^{n} \left( \left\| x_i^t - w_i^t \right\|^2 + \left\| w_i^t - w_i^{t-1} \right\|^2 + \left\| w_i^{t-1} - x_i^{t-1} \right\|^2 \right) \\
&\leq 3n L^2 \eta^2 G^2,
\end{aligned}
$$

where we use $L$-smoothness of the utility function $u_i$ in the first inequality; we use the update rule of OG and the fact that gradients are bounded by $G$ in the last inequality.

Applying the above inequality to the regret bound obtained in Theorem 4, the individual $\Phi_{\text{Proj}}^{\mathcal{X}}(\delta)$-regret of player $i$ is upper bounded by

$$\text{Reg}_{\text{Proj},\delta}^{T,i} \leq \frac{\delta D}{\eta} + \eta G^2 + 3nL^2G^2\eta^3 T.$$

Choosing $\eta = \min\{(\delta D/(nL^2G^2T))^{\frac{1}{4}}, (\delta D)^{\frac{1}{2}}/G\}$, we have $\text{Reg}_{\text{Proj},\delta}^{T,i} \leq 4(\delta D)^{\frac{3}{4}}(nL^2G^2T)^{\frac{1}{4}} + 2\sqrt{\delta D}G$. Using Lemma 1, we have the empirical distribution of played strategy profiles converges to $(\varepsilon, \Phi_{\text{Proj}}(\delta))$-local correlated equilibrium in $O(\max\{\frac{\delta D(nL^2G^2)^{\frac{1}{3}}}{(2\varepsilon-\delta^2 L)^{\frac{4}{3}}}, \frac{\sqrt{\delta D}G}{2\varepsilon-\delta^2 L}\})$ iterations. $\qquad\square$

## D  HARDNESS IN THE GLOBAL REGIME

In the local regime $\delta \leq \sqrt{2\varepsilon/L}$, $(\varepsilon, \delta)$-local Nash equilibrium is intractable and we have shown polynomial-time algorithms for computing the weaker notions of $(\varepsilon, \Phi_{\text{Int}}(\delta))$ and $(\varepsilon, \Phi_{\text{Proj}}(\delta))$-local correlated equilibrium. A natural question is whether correlation enables efficient computation of $(\varepsilon, \Phi(\delta))$-local correlated equilibrium when $\delta$ is in the global regime, i.e., $\delta = \Omega(\sqrt{d})$. In this section, we prove both computational hardness and a query complexity lower bound for both $(\varepsilon, \Phi_{\text{Int}}(\delta))$ and $(\varepsilon, \Phi_{\text{Proj}}(\delta))$-local correlated equilibrium in the global regime

To prove the lower bound results, we only require a single-player game. The problem of computing a $(\varepsilon, \Phi(\delta))$-local correlated equilibrium becomes: given scalars $\varepsilon, \delta, G, L > 0$ and a polynomial-time Turing machine $\mathcal{C}_f$ evaluating a $G$-Lipschitz and $L$-smooth function $f : [0, 1]^d \to [0, 1]$ and its gradient $\nabla f : [0, 1]^d \to \mathbb{R}^d$, we are asked to output a distribution $\sigma$ that is a $(\varepsilon, \Phi(\delta))$-local correlated equilibrium or $\perp$ if such equilibrium does not exist.

**Hardness of $(\varepsilon, \Phi_{\text{Int}}(\delta))$-local correlated equilibrium in the global regime**  When $\delta = \sqrt{d}$, which equals to the diameter $D$ of $[0, 1]^d$, then the problem of finding an $(\varepsilon, \Phi_{\text{Int}}(\delta))$-local correlated equilibrium is equivalent to finding a $(\varepsilon, \delta)$-local minimum of $f$: assume $\sigma$ is an $(\varepsilon, \Phi_{\text{Int}}(\delta))$-local correlated equilibrium of $f$, then there exists $x \in [0, 1]^d$ in the support of $\sigma$ such that

$$f(x) - \min_{x^* \in [0,1]^d} f(x^*) \leq \varepsilon.$$

Then hardness of finding an $(\varepsilon, \Phi_{\text{Int}}(\delta))$-local correlated equilibrium follows from hardness of finding a $(\varepsilon, \delta)$-local minimum of $f$ [DSZ21]. The following Theorem is a corollary of Theorem 10.3 and 10.4 in [DSZ21].

**Theorem 6** (Hardness of $(\varepsilon, \Phi_{\text{Int}}(\delta))$-local correlated equilibrium in the global regime). *In the worst case, the following two holds.*

- *Computing an $(\varepsilon, \Phi_{\text{Int}}(\delta))$-local correlated equilibrium for a game on $\mathcal{X} = [0, 1]^d$ with $G = \sqrt{d}$, $L = d$, $\varepsilon \leq \frac{1}{24}$, $\delta = \sqrt{d}$ is NP-hard.*

- *$\Omega(2^d/d)$ value/gradient queries are needed to determine an $(\varepsilon, \Phi_{\text{Int}}(\delta))$-local correlated equilibrium for a game on $\mathcal{X} = [0, 1]^d$ with $G = \Theta(d^{15})$, $L = \Theta(d^{22})$, $\varepsilon < 1$, $\delta = \sqrt{d}$.*

**Hardness of $(\varepsilon, \Phi_{\text{Proj}}^{\mathcal{X}}(\delta))$-local correlated equilibrium in the global regime**  The proofs of Theorem 7 and Corollary 1 can be found in the next section.

**Theorem 7** (Hardness of $(\varepsilon, \Phi_{\text{Proj}}(\delta))$-local correlated equilibrium in the global regime). *In the worst case, the following two holds.*

- *Computing an $(\varepsilon, \Phi_{\text{Proj}}(\delta))$-local correlated equilibrium for a game on $\mathcal{X} = [0, 1]^d$ with $G = \Theta(d^{15})$, $L = \Theta(d^{22})$, $\varepsilon < 1$, $\delta = \sqrt{d}$ is NP-hard.*

- *$\Omega(2^d/d)$ value/gradient queries are needed to determine an $(\varepsilon, \Phi_{\text{Proj}}(\delta))$-local correlated equilibrium for a game on $\mathcal{X} = [0, 1]^d$ with $G = \Theta(d^{15})$, $L = \Theta(d^{22})$, $\varepsilon < 1$, $\delta = \sqrt{d}$.*

The hardness of computing local correlated equilibrium also implies lower bound on $\Phi_{\text{Proj}}^{\mathcal{X}}(\delta)$-regret in the global regime.

**Corollary 1** (Lower bound of $\Phi_{\text{Proj}}^{\mathcal{X}}(\delta)$-regret against non-convex functions). *In the worst case, the $\Phi_{\text{Proj}}^{\mathcal{X}}(\delta)$-regret of any online algorithm is at least $\Omega(2^d/d, T)$ even for loss functions $f : [0,1]^d \to [0,1]$ with $G, L = \text{poly}(d)$ and $\delta = \sqrt{d}$.*

### D.1 PROOF OF THEOREM 7

We will reduce the problem of finding $(\varepsilon, \Phi_{\text{Proj}}(\delta))$-local correlated equilibrium in smooth games to finding a satisfying assignment of a boolean function, which is NP-complete.

**Fact 1.** *Given only black-box access to a boolean formula $\phi : \{0,1\}^d \to \{0,1\}$, at least $\Omega(2^d)$ queries are needed in order to determine whether $\phi$ admits a satisfying assignment $x^*$ such that $\phi(x^*) = 1$. The term* black-box access *refers to the fact that the clauses of the formula are not given and the only way to determine whether a specific boolean assignment is a satisfying is by querying the specific binary string. Moreover, the problem of finding a satisfying assigment of a general boolean function is NP-hard.*

We revisit the construction of the hard instance in the proof of [DSZ21, Theorem 10.4] and use its specific structures. Given black-box access to a boolean formula $\phi$ as described in Fact 1, following [DSZ21], we construct the function $f_\phi(x) : [0,1]^d \to [0,1]$ as follows:

1. for each corner $v \in V = \{0,1\}^d$ of the $[0,1]^d$ hypercube, we set $f_\phi(x) = 1 - \phi(x)$.

2. for the rest of the points $x \in [0,1]^d/V$, we set $f_\phi(x) = \sum_{v \in V} P_v(x) \cdot f_\phi(v)$ where $P_v(x)$ are non-negative coefficients defined in [DSZ21, Definition 8.9].

The function $f_\phi$ satisfies the following properties:

1. if $\phi$ is not satisfiable, then $f_\phi(x) = 1$ for all $x \in [0,1]^d$ since $f_\phi(v) = 1$ for all $v \in V$; if $\phi$ has a satisfying assignment $v^*$, then $f_\phi(v^*) = 0$.

2. $f_\phi$ is $\Theta(d^{12})$-Lipschitz and $\Theta(d^{25})$-smooth.

3. for any point $x \in [0,1]^d$, the set $V(x) := \{v \in V : P_v(x) \neq 0\}$ has cardinality at most $d+1$ while $\sum_{v \in V} P_v(x) = 1$; any value / gradient query of $f_\phi$ can be simulated by $d+1$ queries on $\phi$.

In the case there exists a satisfying argument $v^*$, then $f_\phi(v^*) = 0$. Define the deviation $e$ so that $e[i] = 1$ if $v^*[i] = 0$ and $e[i] = -1$ if $v^*[i] = 1$. It is clear that $\|e\| = \sqrt{d} = \delta$. By properties of projection on $[0,1]^d$, for any $x \in [0,1]^d$, we have $\Pi_{[0,1]^n}[x-v] = v^*$. Then any $(\varepsilon, \Phi_{\text{Proj}}(\delta))$-local correlated equilibrium $\sigma$ must include some $x^* \in \mathcal{X}$ with $f_\phi(x^*) < 1$ in the support, since $\varepsilon < 1$. In case there exists an algorithm $\mathcal{A}$ that computes an $(\varepsilon, \Phi_{\text{Proj}}(\delta))$-local correlated equilibrium, $\mathcal{A}$ must have queried some $x^*$ with $f_\phi(x^*) < 1$. Since $f_\phi(x^*) = \sum_{v \in V(x^*)} P_v(x^*) f_\phi(v) < 1$, there exists $\hat{v} \in V(x^*)$ such that $f_\phi(\hat{v}) = 0$. Since $|V(x^*)| \leq d+1$, it takes addition $d+1$ queries to find $\hat{v}$ with $f_\phi(\hat{v}) = 0$. By fact 1 and the fact that we can simulate every value / gradient query of $f_\phi$ by $d+1$ queries on $\phi$, $\mathcal{A}$ makes at least $\Omega(2^d/d)$ value / gradient queries.

Suppose there exists an algorithm $\mathcal{B}$ that outputs an $(\varepsilon, \Phi_{\text{Proj}}(\delta))$-local correlated equilibrium $\sigma$ in time $T(\mathcal{B})$ for $\varepsilon < 1$ and $\delta = \sqrt{d}$. We construct another algorithm $\mathcal{C}$ for SAT that terminates in time $T(\mathcal{B}) \cdot \text{poly}(d)$. $\mathcal{C}$: (1) given a boolean formula $\phi$, construct $f_\phi$ as described above; (2) run $\mathcal{B}$ and get output $\sigma$ (3) check the support of $\sigma$ to find $v \in \{0,1\}^d$ such that $f_\phi(v) = 0$; (3) if finds $v \in \{0,1\}^d$ such that $f_\phi(v) = 0$, then $\phi$ is satisfiable, otherwise $\phi$ is not satisfiable. Since we can evaluate $f_\phi$ and $\nabla f_\phi$ in $\text{poly}(d)$ time and the support of $\sigma$ is smaller than $T(\mathcal{B})$, the algorithm $\mathcal{C}$ terminates in time $O(T(\mathcal{B}) \cdot \text{poly}(d))$. The above gives a polynomial time reduction from SAT to $(\varepsilon, \Phi_{\text{Proj}}(\delta))$-local correlated equilibrium and proves the NP-hardness of the latter problem.

### D.2 PROOF OF COROLLARY 1

Let $\phi : \{0,1\}^d \to \{0,1\}$ be a boolean formula and define $f_\phi : [0,1]^d \to [0,1]$ the same as that in Theorem 7. We know $f_\phi$ is $\Theta(\text{poly}(d))$-Lipschitz and $\Theta(\text{poly}(d))$-smooth. Now we let the adversary picks $f_\phi$ in each time. For any $T \leq O(2^d/d)$, in case there exists an online learning

algorithm with $\mathrm{Reg}_{\mathrm{Proj},\delta}^T < \frac{T}{2}$, then $\sigma := \frac{1}{T}\sum_{t=1}^T 1_{x^t}$ is an $(\frac{1}{2},\delta)$-local correlated equilibrium. Applying Theorem 7 and the fact that in this case $\mathrm{Reg}_{\mathrm{Proj},\delta}^T$ is non-decreasing with respect to $T$ concludes the proof.

## E    REMOVING THE $D$ DEPENDENCE

For the regime $\delta \le D_{\mathcal{X}}$ which we are more interested in, the lower bound in Theorem 3 is $\Omega(G\delta\sqrt{T})$ while the upper bound in Theorem 2 is $O(G\sqrt{\delta D_{\mathcal{X}} T})$. They are not tight especially when $D_{\mathcal{X}} \gg \delta$. A natural question is: *which of them is the tight bound?* We conjecture that the lower bound is tight. In fact, for the special case where the feasible set $\mathcal{X}$ is a *box*, we have a way to obtain a $D_{\mathcal{X}}$-independent bound $O(d^{\frac{1}{4}}G\delta\sqrt{T})$, which is tight when $d=1$. Below, we first describe the improved strategy in 1-dimension. Then we show how to extend it to the $d$-dimensional box setting.

### E.1    ONE-DIMENSIONAL CASE

In one-dimension, we assume that $\mathcal{X} = [a,b]$ for some $b - a \ge 2\delta$ (if $b - a \le 2\delta$, then our original bound in Theorem 2 is already of order $G\delta\sqrt{T}$). We first investigate the case where $f^t(x)$ is a linear function, i.e., $f^t(x) = g^t x$ for some $g^t \in [-G, G]$. The key idea is that we will only select $x^t$ from the two intervals $[a, a+\delta]$ and $[b-\delta, b]$, and never play $x^t \in (a+\delta, b-\delta)$. To achieve so, we concatenate these two intervals, and run an algorithm in this region whose diameter is only $2\delta$. The key property we would like to show is that the regret is preserved in this modified problem.

More precisely, given the original feasible set $\mathcal{X} = [a,b]$, we create a new feasible set $\mathcal{Y} = [-\delta, \delta]$ and apply our algorithm GD in this new feasible set. The loss function is kept as $f^t(x) = g^t x$. Whenever the algorithm for $\mathcal{Y}$ outputs $y^t \in [-\delta, 0]$, we play $x^t = y^t + a + \delta$ in $\mathcal{X}$; whenever it outputs $y^t \in (0, \delta]$, we play $x^t = y^t + b - \delta$. Below we show that the regret is the same in these two problems. Notice that when $y^t \le 0$, we have for any $v \in [-\delta, \delta]$,

$$
\begin{aligned}
x^t - \Pi_{\mathcal{X}}[x^t - v] &= x^t - \max\left(\min\left(x^t - v, b\right), a\right) \\
&= x^t - \max\left(x^t - v, a\right) \\
&\qquad\qquad (x^t - v = y^t + a + \delta - v \le a + 2\delta \le b \text{ always holds}) \\
&= y^t + a + \delta - \max\left(y^t + a + \delta - v, a\right) \\
&= y^t - \max\left(y^t - v, -\delta\right) \\
&= y^t - \max\left(\min\left(y^t - v, \delta\right), -\delta\right) \qquad (y^t - v \le \delta \text{ always holds}) \\
&= y^t - \Pi_{\mathcal{Y}}[y^t - v]
\end{aligned}
$$

Similarly, when $y^t > 0$, we can follow the same calculation and prove $x^t - \Pi_{\mathcal{X}}[x^t - v] = y^t - \Pi_{\mathcal{Y}}[y^t - v]$. Thus, the regret in the two problems:

$$
g^t\left(x^t - \Pi_{\mathcal{X}}[x^t - v]\right) \quad \text{and} \quad g^t\left(y^t - \Pi_{\mathcal{Y}}[y^t - v]\right)
$$

are exactly the same for any $v$. Finally, observe that the diameter of $\mathcal{Y}$ is only of order $O(\delta)$. Thus, the upper bound in Theorem 2 would give us an upper bound of $O(G\sqrt{\delta \cdot \delta T}) = O(G\delta\sqrt{T})$.

For convex $f^t$, we run the algorithm above with $g^t = \nabla f^t(x^t)$. Then by convexity we have

$$
f^t(x^t) - f^t(\Pi_{\mathcal{X}}[x^t - v]) \le g^t(x^t - \Pi_{\mathcal{X}}[x^t - v]) = g^t(y^t - \Pi_{\mathcal{Y}}[y^t - v]),
$$

so the regret in the modified problem (which is $O(G\delta\sqrt{T})$) still serves as a regret upper bound for the original problem.

### E.2    $d$-DIMENSIONAL BOX CASE

A $d$-dimensional box is of the form $\mathcal{X} = [a_1, b_1] \times [a_2, b_2] \times \cdots \times [a_d, b_d]$. The box case is easy to deal with because we can decompose the regret into individual components in each dimension.

Namely, we have

$$f^t(x^t) - f^t(\Pi_{\mathcal{X}}[x^t - v]) \leq \nabla f^t(x^t)^\top \left( x^t - \Pi_{\mathcal{X}}[x^t - v] \right)$$

$$= \sum_{i=1}^{d} g_i^t \left( x_i^t - \Pi_{\mathcal{X}_i}[x_i^t - v_i] \right)$$

where we define $\mathcal{X}_i = [a_i, b_i]$, $g^t = \nabla f^t(x^t)$, and use subscript $i$ to indicate the $i$-th component of a vector. The last equality above is guaranteed by the box structure. This decomposition allows as to view the problem as $d$ independent 1-dimensional problems.

Now we follow the strategy described in Section E.1 to deal with individual dimensions (if $b_i - a_i < 2\delta$ then we do not modify $\mathcal{X}_i$; otherwise, we shrink $\mathcal{X}_i$ to be of length $2\delta$). Applying the analysis of Theorem 2 to each dimension, we get

$$\sum_{i=1}^{d} g_i^t \left( x_i^t - \Pi_{\mathcal{X}_i}[x_i^t - v_i] \right)$$

$$\leq \sum_{i=1}^{d} \left( \frac{v_i^2}{2\eta} + \frac{\eta}{2} \sum_{t=1}^{T} (g_i^t)^2 + \frac{|v_i| \times 2\delta}{\eta} \right) \quad \text{(the diameter in each dimension is now bounded by } 2\delta\text{)}$$

$$\leq O\left( \frac{\delta \sum_{i=1}^{d} |v_i|}{\eta} + \eta G^2 T \right)$$

$$\leq O\left( \frac{\delta^2 \sqrt{d}}{\eta} + \eta G^2 T \right). \quad \text{(by Cauchy-Schwarz, } \sum_i |v_i| \leq \sqrt{d}\sqrt{\sum_i |v_i|^2} \leq \delta\sqrt{d}\text{)}$$

Choosing the optimal $\eta = \frac{d^{\frac{1}{4}}\delta}{G\sqrt{T}}$, we get the regret upper bound of order $O\left( d^{\frac{1}{4}} G\delta\sqrt{T} \right)$.

