# OpenReview forum: "On Local Equilibrium in Non-Concave Games"
_ICLR.cc/2024/Conference — Submitted to ICLR 2024_

### Official Review · Reviewer_J78b · 2023-10-20

**Soundness:** 3 good
**Presentation:** 2 fair
**Contribution:** 1 poor
**Rating:** 3
**Confidence:** 3

**Summary:**

The authors propose a notion of “local correlated equilibrium” for non-concave games, and show that variants of GD converge to this solution concept.

**Strengths:**

Some of the algorithms require careful analysis? I'm not sure...

**Weaknesses:**

At a high level, I’m concerned about the motivation. The authors introduce a new solution concept and design algorithms, but don’t really stop to motivate their solution concept. The way I understand, in practice GAN training with OGD has limited success because it gets stuck in cycles. Now you’re basically telling me that the path of this training satisfies some new solution concept. What should I learn from that? By analogy, in Game Theory correlated equilibrium has a natural interpretation with a correlating device, and is known to satisfy some good properties (“Price of Anarchy”). What can I do with the fact that the trajectory of my GAN training algorithm is an approximate “local correlated equilibrium”?

**Questions:**

[These are more writing comments - but feel free to answer my questions from "weaknesses" section]



The paper is motivated by a hardness result from [DSZ21] for the stronger notion of local Nash equilibrium. But the hardness result in [DSZ21] holds *only* in a non-standard setting where the feasible domain is not a product. In contrast, your work seems to rely on having a product domain.


I think your solution concept should be called “local *coarse* correlated equilibrium”: You consider a single deviation rule and want to apply it to all x’s in the distribution. This also explains why you can find it by minimizing external regret.

The title should absolutely be updated to say something about (coarse) correlated.


There are two definitions (2 and 4) called “local correlated equilibrium”

---

> ### Author Response · Authors · 2023-11-18
> **Response to Reviewer J78b-Part 1**
>
> **Q**: *At a high level, I’m concerned about the motivation. The authors introduce a new solution concept and design algorithms, but don’t really stop to motivate their solution concept. The way I understand, in practice GAN training with OGD has limited success because it gets stuck in cycles. Now you’re basically telling me that the path of this training satisfies some new solution concept. What should I learn from that? By analogy, in Game Theory correlated equilibrium has a natural interpretation with a correlating device, and is known to satisfy some good properties (“Price of Anarchy”). What can I do with the fact that the trajectory of my GAN training algorithm is an approximate “local correlated equilibrium”?*
>
> **A**: Thank you for the comment on the motivation of our proposed solution concepts. We appreciate the opportunity to further elucidate our goals. Our primary objective is to introduce a new solution concept for non-concave games that satisfies  the following three requirements: game theoretic meaningfulness, universality, and computational tractability. Our notion of local correlated equilibrium is meaningful as it generalizes the notion of correlated equilibrium and coarse correlated equilibrium to non-concave games. Additionally, as we argue in the paper, this solution concept always exists, hence its universality. Finally, for the two instantiations, we show that these solution concepts are computationally tractable.
>
> As highlighted by the reviewer, correlated equilibrium is a useful concept when there is a signaling device that could be used to correlate different agents’ actions. However, in the context of non-concave games, finding a correlated equilibrium is computationally intractable. We believe our solution concept serves as a useful alternative in these non-concave games where a signaling device is available, which includes many multi-agent systems such as market dynamics (see the respones to Review oLqg (Q1) for an example), cooperative strategies among self-driving cars, and interactions within social networks.
>
> Finally, it’s important to note that the training GANs is a min-max optimization problem where the goal is to identify the minimax strategy for the min-player (controlling the generator network). Our focus on finding a local correlated equilibrium is very different from the pursuit of a minimax strategy, and as such, doesn’t inherently guarantee high performance of the generator.
>
> **Q**: *The paper is motivated by a hardness result from [DSZ21] for the stronger notion of local Nash equilibrium. But the hardness result in [DSZ21] holds only in a non-standard setting where the feasible domain is not a product. In contrast, your work seems to rely on having a product domain.*
>
> **A**: Thanks for pointing this out! The result from [DSZ21] shows that local Nash equilibrium is intractable even in two-player **zero-sum** games with coupled domain. Thus whether local Nash equilibrium in two-player zero-sum games can be efficiently computed with product domain is still open. However, we study the more general multi-player non-concave games that include two-player **general-sum** games as a special case. Note that in a two-player general-sum bimatrix game, since the loss function is linear in each player’s strategy, it is not hard to see that every local Nash equilibrium is also a Nash equilibrium, and, therefore, is PPAD-hard to compute [DGP09, CDT09]. The intractability of local Nash equilibrium in general-sum games motivates our new local solution concepts that is universal for multiplayer non-concave smooth games.
>
> [DSZ21] Daskalakis, Constantinos, Stratis Skoulakis, and Manolis Zampetakis. "The complexity of constrained min-max optimization." STOC, 2021
>
> [DGP09] Daskalakis, Constantinos, Paul W. Goldberg, and Christos H. Papadimitriou. "The complexity of computing a Nash equilibrium." SICOMP, 2009
>
> [CDT09] Chen, Xi, Xiaotie Deng, and Shang-Hua Teng. "Settling the complexity of computing two-player Nash equilibria." JACM, 2009

---

> > ### Comment · Reviewer_J78b · 2023-11-20
> > **DZS vs DGP vs CDT**
> >
> > Your current version is still citing the hardness result of DSZ without mentioning that it doesn't apply at all to the setting you're considering.
> >
> > Note that DGP only proved hardness against 1/exp(n)-approximate Nash equilibrium, while your regret-based algorithms only give 1/poly-approximate local \Phi-correlated equilibrium, so it is not particularly relevant.
> >
> > You are right that CDT proved hardness of 1/poly-approximate Nash equilibrium which trivially extends from bilinear to general non-concave games. But you don't cite CDT at all in your paper!

---

> > > ### Author Response · Authors · 2023-11-21
> > > **Response to "DZS vs DGP vs CDT"**
> > >
> > > We have added a revised discussion on the hardness of computing local Nash equilibria, along with a reference to [CDT09].

---

> ### Author Response · Authors · 2023-11-18
> **Response to Reviewer J78b-Part 2**
>
> **Q**: *I think your solution concept should be called “local coarse correlated equilibrium”: You consider a single deviation rule and want to apply it to all x’s in the distribution. This also explains why you can find it by minimizing external regret.*
>
> **A**: We would like to clarify that both coarse correlated equilibrium (CCE) and correlated equilibrium (CE) consider a single strategy modification (deviation rule) that is applied to all strategies. The core difference between CCE and CE is that CCE allows only constant strategy modification that maps any strategy to a fixed strategy, while CE allows all possible linear transformations over the strategy space (i.e., the simplex). In the context of normal-form games, both CCE and CE are special cases of $\Phi$-equilibria: a CCE is a $\Phi_{ext}$-equilibrium where $\Phi_{ext}$ contains only constant strategy modifications; a CE is a $\Phi_{swap}$-equilibrium where $\Phi_{swap}$ contains all linear transformations $\phi: \Delta \rightarrow \Delta$.
>
> The solution concept we propose, $(\epsilon, \Phi(\delta))$-local correlated equilibrium, is determined by the set of strategy modifications $\Phi(\delta)$. In general, $\Phi(\delta)$ could include all possible (local) strategy modifications, not only constant strategy modifications. The two specific instantiations considered in the paper, $\Phi_{int}(\delta)$ and $\Phi_{proj}(\delta)$, also allow non-constant strategy modifications. That is why we choose to call it local correlated equilibrium instead of local coarse correlated equilibrium. Additionally, we have showed in the paper that $Reg_{proj}(\delta)$ is incomparable with the external regret and minimizing external regret does not always lead to a $\Phi_{proj}(\delta)$-local correlated equilibrium (Example 1 and 2 in section 5).
>
> **Q**: There are two definitions (2 and 4) called “local correlated equilibrium”
>
> **A**: Thanks for pointing out! We have changed Definition 4 to "Two Instantiations of Local Correlated Equilibrium".

---

> > ### Comment · Reviewer_J78b · 2023-11-20
> > **Correlated vs Coarse Correlated equilibrium**
> >
> > I agree that for certain choices of $\Phi$ you *could* get something that could be honestly called local correlated equilibrium.
> >
> > But the choices of $\Phi$ that you actually analyze in your paper, while they don't perfectly align with either classical notions of external or internal, are much closer in spirit to external regret in the sense that the allowed transformations are defined by a single strategy in the domain. I guess you could call it local $\Phi$-correlated or something like that. But just "local correlated" is inaccurate, borderline misleading.

---

> > > ### Author Response · Authors · 2023-11-21
> > > **Response to "Correlated vs Coarse Correlated equilibrium"**
> > >
> > > **Q**: *I agree that for certain choices of $\Phi$ you could get something that could be honestly called local correlated equilibrium. But the choices of $\Phi$ that you actually analyze in your paper, while they don't perfectly align with either classical notions of external or internal, are much closer in spirit to external regret in the sense that the allowed transformations are defined by a single strategy in the domain. I guess you could call it local $\Phi$-correlated or something like that. But just "local correlated" is inaccurate, borderline misleading.*
> > >
> > > **A**: Formally, our notion is called $(\epsilon, \Phi(\delta))$-local correlated equilibrium (Definition 2), which is parameterized by $\epsilon$ and the set $\Phi(\delta)$. In particular, the two instantiations we analyze are $(\epsilon, \Phi_{int}(\delta))$-local correlated equilibrium and $(\epsilon, \Phi_{proj}(\delta))$-local correlated equilibrium (Definition 4). We agree with the reviewer that these two instantiations are closer in spirit to coarse correlated equilibrium. We have revised our paper to consistently specify our choice of $\Phi$ when discussing this solution concept.

---

### Official Review · Reviewer_oLqg · 2023-11-01

**Soundness:** 4 excellent
**Presentation:** 3 good
**Contribution:** 3 good
**Rating:** 8
**Confidence:** 3

**Summary:**

The authors try to shed light on a new chapter of algorithmic game theory -- i.e., nonconcave games. Nonconcave games are simply games where the utility function of each player is nonconcave with respect to their individual strategy.

Such games have come to the attention of theoreticians due to the advent of an array of machine learning applications. Traditional notions of individual rationality such as the Nash equilibrium need not exist in these games while relaxed notions of equilibria designed for nonconvex games can be intractable. Namely, local $\epsilon$-approximate Nash equilibria is a strategy profile in which no agent can improve their utility more than $\epsilon$ by only considering strategy deviations of distance $\delta$ from the initial strategy. Yet, $(\epsilon, \delta)$-local NE are either trivial to compute, PPAD-hard, or NP-hard (corresponding to the magnitude of $\delta$ compared to the natural parameters of the game). The latter two cases are known as the *local* and the *global regime*.

To this end, the authors propose the notion of a *local correlated equilibrium* as to alleviate the intractability of local-NE in the local regime. After they define this new notion of equilibrium they review the notion of $\Phi$-regret. Briefly, $\Phi$-regret unifies various notions of regret (e.g., external regret, swap regret) under an umbrella definition; it is defined as the difference between in utility at the end of the online optimization process where the best strategy in hindsight is selected using a family of function $\Phi$.

The latter notion is crucial not only for the purpose of an algorithmic solution as well as the notion of the equilibrium itself. An $(\epsilon, \Phi(\delta))$-correlated equilibrium is roughly a correlated strategy profile that achieves small $\Phi(\delta)$-regret for each agent. $\Phi(\delta)$-regret is the $\Phi$-regret where the family of modification functions only allow deviations in a radius of length $\delta$.

The authors note that, to date, there does not exist an efficient algorithm for $\Phi$-regret minimization for general sets $\Phi$. As such, two families of $\Phi$ are considered:
* Interpolations between current strategies from fixed strategies
* Deviations towards a given direction $v$ in a distance of length $\delta$.

Then, the authors utilize the existing online convex optimization framework (the gradient descent and optimistic gradient descent algorithms) to straightforwardly design algorithms that lead to $(\epsilon, \Phi(\delta))$-correlated equilibria.

As a takeaway, the authors propose that solution concepts in nonconcave games should be *meaningful, universal, and tractable*.  I suspect these notions would take the place of rationality. Nevertheless, there is not an explicit discussion as to why their proposal attains these favorable properties.

**Strengths:**

* The motivation is clear and is guided by both existing applications and contemporary theoretical advances.
* The paper introduces algorithmic solutions and equilibrium concepts for a nascent family of games that arguably can be proven of great importance in the future.
* The algorithmic framework is quite versatile and able to fit different instances of no-regret algorithms and $\Phi$ function families.
* The computational complexity issues are discussed and explained with clarity.

**Weaknesses:**

* One has to be fair and recognize the novelty of the paper and the absence of pre-existing criteria for its assessment; nevertheless, it would be rational to ask for some justification of the proposed equilibrium notion other than computational complexity arguments. In a sense, what are real-world examples where the proposed notions of equilibria are already established as desirable states of a game?

* A more precise meaning of what a meaningful and universal equilibrium is remains unclear from the text. It would be nice if the authors could elaborate on those concepts and what makes the particular $\epsilon, \Phi(\delta)$-correlated equilibria attain these properties.

**Questions:**

* What kinds of $\Phi(\delta)$ families would the authors consider as important for future study and of game-theoretic importance?
* What is the connection of $\Phi(\delta)$-regret minimization and bounded rationality? Putting the computational theoretic aspects aside, we in a sense assume agents to be as rational as their first-order derivative dictates. Would assuming bounded rationality for the agents lead to tractable notions of equilibria as well?
* What would qualitatively change if we assumed access to second-order information?

---

> ### Author Response · Authors · 2023-11-18
> **Response to Reviewer oLqg-Part 1**
>
> Thank you for your very detailed review and insightful questions! We address your questions below
>
> **Q1**: *One has to be fair and recognize the novelty of the paper and the absence of pre-existing criteria for its assessment; nevertheless, it would be rational to ask for some justification of the proposed equilibrium notion other than computational complexity arguments. In a sense, what are real-world examples where the proposed notions of equilibria are already established as desirable states of a game?*
>
> **A**: We appreciate your recognition of the novelty of our work and your interest in the practical relevance of our proposed equilibrium notion. We will use the dynamics of pricing in a competitive market as an example to illustrate the applicability of our $\Phi_{proj}(\delta)$-local correlated equilibrium concept in a real-world scenario.
>
> The market has multiple agents selling identical goods. Each agent sets a price $p^i_t$ for their goods at day $t >=1$. The reward everyday depends on the agent’s price: a high price decreases the demand of the goods but increases revenue per unit; while a low price could increase demand but decreases revenue per unit. In a complex market environment, the utility function in terms of $p_t^i$ could be non-concave so that even computing the optimal price given the knowledge of the utility functions is intractable. If each player adjusts their price using the online gradient descent algorithm, then our results show that their pricing strategy converges to a $\Phi_{proj}(\delta)$-local correlated equilibrium. In such equilibrium, each player would not regret to be more conservative (i.e., posting a slightly lower price $p_t- \delta$ everyday) or to be more aggressive (i.e., posting a slightly higher price $p_t + \delta$ everyday). This scenario demonstrates the equilibrium's relevance: in a market where agents have bounded rationality and make decisions based on localized price adjustments, our proposed equilibrium notion provides a stability guarantee. More generally, the proposed solution concepts may have practical implications, especially in multi-agent systems where agents encounter complex, non-concave utility functions and operate under the constraints of bounded rationality.
>
> **Q2**: *What kinds of $\Phi(\delta)$ families would the authors consider as important for future study and of game-theoretic importance?*
>
> A: This is a great question!  One family of $\Phi(\delta)$ that is important is a generalization of $\Phi_{int}(\delta)$. Recall that $\Phi_{int}(\delta)$ contains strategy modifications that interpolates between current strategy $x$ and a fixed strategy $x^*$. One can consider a strategy modification that interpolates between current strategy $x$ and a modified strategy $\phi(x)$ for $\phi \in \Phi’$ where $\Phi'$ is a set of (possibly not local) strategy modifications. $\Phi_{int}(\delta)$ then becomes a special case where $\phi(x) = x^*$ is a constant strategy modification. By a similar proof to that in section 4, it is easy to show that regret minimization under the new set reduces to $\Phi’$-regret minimization. We think obtaining a more fine-grained understanding of the complexity of $\Phi’$-regret minimization for more general classes of strategy modifications $\Phi’$ is an interesting future direction. For instance, is it possible to define a meaningful complexity measure of a set $\Phi’$ that characterizes the computational / oracle complexity of $\Phi’$-regret minimization.

---

> > ### Comment · Reviewer_J78b · 2023-11-20
> > **Confused about authors' reply to Q1: why not just focus on regret?**
> >
> > I'm confused about your response to this reviewer's Q1 about the (lack) of motivation for the new solution concept that you introduce, which you also cited in your response to my review:
> > You propose to motivate your solution concept in that if the no-regret dynamics converge to it, then each player would have low regret. But if all you're trying to achieve is low regret, why talk about equilibria at all? I.e. why not just analyze the performance of no-regret algorithms in non-concave games?

---

> ### Author Response · Authors · 2023-11-18
> **Response to Reviewer oLqg-Part 2**
>
> **Q3**: *What is the connection of $\Phi(\delta)$-regret minimization and bounded rationality? Putting the computational theoretic aspects aside, we in a sense assume agents to be as rational as their first-order derivative dictates. Would assuming bounded rationality for the agents lead to tractable notions of equilibria as well?*
>
> **A**: This is a great question! We believe $\Phi(\delta)$-regret minimization is a model of  bounded rationality. In this context, we refer to bounded rationality as the idea that agents have limitations in their decision-making processes, particularly in terms of the information they consider and the computational resources they have.
>
> In $\Phi(\delta)$-regret minimization, the set $\Phi(\delta)$ only contains local deviations that are bounded by $\delta$. Thus agents are not fully rational as they ignore global deviations that might be more beneficial. Moreover, as you pointed out, assuming agents to be as rational as their first-order derivative dictates is a choice of bounded rationality. Furthermore, the cardinality and complexity of the set of $\Phi(\delta)$ reflect a degree of an agent’s rationality: A more rational agent, within the bounds of their capabilities, might consider a larger and more complex set of potential deviations compared to a less rational agent. Our results demonstrate that two sets of $\Phi(\delta)$ lead to tractable notions of local correlated equilibrium. This suggests that by assuming bounded rationality, agents can achieve stable outcomes even in complex non-concave environments.
>
> Exploring other models of bounded rationality, different equilibrium concepts, and learning dynamics that converge to these equilibria are very interesting future directions.
>
> **Q4**: *What would qualitatively change if we assumed access to second-order information?*
>
> **A**: We remark that by assuming access to second-order information, [DGSZ23] developed a second-order algorithm that converges to local Nash equilibrium although without concrete convergence rates. This result improves over first-order methods which usually get stuck in limit cycles and shows the power of second-order methods. If we assume access to second-order information, it is possible that we can develop algorithms with faster convergence to local correlated equilibrium than first-order algorithms. This is a very interesting future direction.
>
> [DGSZ23] Daskalakis, C., Golowich, N., Skoulakis, S., & Zampetakis, E. STay-ON-the-Ridge: Guaranteed Convergence to Local Minimax Equilibrium in Nonconvex-Nonconcave Games. COLT, 2023

---

> ### Author Response · Authors · 2023-11-21
> **Response to "Confused about authors' reply to Q1: why not just focus on regret?"**
>
> **Q**: *I'm confused about your response to this reviewer's Q1 about the (lack) of motivation for the new solution concept that you introduce, which you also cited in your response to my review: You propose to motivate your solution concept in that if the no-regret dynamics converge to it, then each player would have low regret. But if all you're trying to achieve is low regret, why talk about equilibria at all? I.e. why not just analyze the performance of no-regret algorithms in non-concave games?*
>
> **A**: There seems to be some misunderstanding regarding the example's purpose. It was intended to illustrate the kind of guarantee our solution concept offers. To rephrase: If a correlating device (mediator) suggests prices to each player according to an $(\epsilon, \Phi_{proj}(\delta))$-local correlated equilibrium, then no player would have more than $\epsilon$-incentive to deviate slightly from the recommended price, either by increasing or decreasing it.  This example serves to motivate the  $(\epsilon, \Phi_{proj}(\delta))$-local correlated equilibrium by showing its ability to provide stability guarantees under a certain form of bounded rationality assumption.
>
> An additional feature of our solution concept is that it is computationally tractable. Indeed, they can be computed if all players employ simple no-regret learning algorithms. Guaranteed convergence by decentralized learning dynamics is an appealing feature of our solution concept, and this property holds for (coarse) correlated equilibrium in concave games and for Nash equilibrium in two-player zero-sum bimatrix games.

---

> > ### Comment · Reviewer_J78b · 2023-11-21
> >
> > Correction: no player would have an incentive to apply a *fixed* deviation that increases/decreases the recommended price by the same amount regardless of the recommendation.
> >
> > 1. I think your paper desparately needs to include a discussion of such a motivating example
> >
> > 2. It would help to motivate the computational complexity question if the motivating example is high dimensional

---

### Official Review · Reviewer_kZP2 · 2023-11-05

**Soundness:** 4 excellent
**Presentation:** 3 good
**Contribution:** 4 excellent
**Rating:** 10
**Confidence:** 4

**Summary:**

This paper proposes a new solution concept called $\phi$-local correlated equilibrium for non-concave games with smooth utilities. The authors show that this concept captures the convergence guarantees of Online Gradient Descent and no-regret learning in such games for two specific initializations of $\phi$. They also provide a new algorithm for computing local correlated equilibria that is based on a variant of Online Gradient Descent. The paper concludes with experimental results that demonstrate the effectiveness of this algorithm in practice.

**Strengths:**

The paper provides important mathematical characterizations for the limit point of multiagent learning algorithms in non-convex game settings and answers important open question posed by Daslakakis et al. [1]

**Weaknesses:**

This is relatively minor but the organization of the paper in my opinion makes the paper hard to read. A few suggestions:
Adding a mathematical description of the problem (i.e., games) to the introduction
Moving some parts of the local correlated equilibrium section on page 2 above the contributions section and tie it in with this mathematical description
Adding more intuition and background on intractability of approximate local Nash to intro together with a mathematical description

**Questions:**

Minor comments and questions:
Aren’t part 1) of assumption 1 redundant given part 3? And part 2) redundant given part 1 and compactness of strategy sets?

The local Nash definition that is studied in the paper considers only *pure* strategies, however, local correlated equilibrium is studied in correlated **mixed** strategies (logically). This begs the questions, can mixed local Nash equilibria be efficiently computed or is that out of reach as well? It seems like that would be out of reach since the randomization would reduce the problem to a multilinear game (albeit infinite dimensional) for which computation of Nash is PPAD. I think a description of this point is important to understand the jump from pure strategies to mixed strategies

Does Lemma 1 assume Lipschitz smoothness/continuity on the convex regrets or no?

How does part 2 of Lemma 1 relate to Hazan et al’s [2] results and in general how do the authors’ result relate to your results on projected \phi regret ?


Naive regret bound in section 3.1 seems meaninglessly loose. That is, having an additive Lipschitz continuity constant G suggests that the algorithm might make no progress at all?


Reg_proj does not have a learning rate in the step it takes this seems to affect the notion that projected and external regret can in general be unrelated?


Writing: For large enough δ, Definition 1 captures global Nash equilibrium as well >> For large enough δ, Definition 1 captures global $\varepsilon$-Nash equilibrium as well


I would love to hear answer to my questions above, but otherwise I think the authors have written an interesting and illuminating paper which deserves acceptance.





[1] Daskalakis, Constantinos, Stratis Skoulakis, and Manolis Zampetakis. "The complexity of constrained min-max optimization." Proceedings of the 53rd Annual ACM SIGACT Symposium on Theory of Computing. 2021.

[2] Hazan, Elad, Karan Singh, and Cyril Zhang. "Efficient regret minimization in non-convex games." International Conference on Machine Learning. PMLR, 2017.

---

> ### Author Response · Authors · 2023-11-18
> **Response to Reviewer kZP2-Part 1**
>
> Thank you for your very positive review and constructive comments! We will incorporate your suggestions in the next version of the paper. We address the questions below.
>
> **Q1**: *Aren’t part 1) of assumption 1 redundant given part 3? And part 2) redundant given part 1 and compactness of strategy sets?*
>
> **A**: Thank you for your thoughtful observations regarding the structure of Assumption 1.
>
> Regarding your first point, we agree that Part 1 of Assumption 1 is implied by Part 3. Part 1 should be the definition of differentiable games while smooth games are differentiable games with additional smoothness assumptions. For clarity, we have removed Part 1 and added the definition of differentiable games in the paragraph above.
>
> Concerning Part 2, your insight is correct that it could be inferred from the combination of the former Part 1 and the compactness of strategy sets. Nevertheless, we have chosen to retain this part for explicitness, particularly to specify the Lipschitz constant $G$. This explicit specification is useful for some of our subsequent analyses and results. We have rephrased it in the revised version of the paper to clarify its role.
>
> These changes, along with the revised introduction of differentiable and smooth games, are highlighted in the updated version of our paper.
>
> **Q2**: *The local Nash definition that is studied in the paper considers only pure strategies, however, local correlated equilibrium is studied in correlated mixed strategies (logically). This begs the questions, can mixed local Nash equilibria be efficiently computed or is that out of reach as well? It seems like that would be out of reach since the randomization would reduce the problem to a multilinear game (albeit infinite dimensional) for which computation of Nash is PPAD. I think a description of this point is important to understand the jump from pure strategies to mixed strategies*
>
> **A**: Thank you for your insightful question on the complexity of mixed local Nash equilibria, which indeed warrants further discussion in our paper.
>
> To address your question: Computing mixed local Nash equilibria is PPAD-hard for general-sum normal-form games, which are concave games. Since the agents have multi-linear utility functions, every local mixed Nash equilibrium is also a Nash equilibrium. Computing a Nash equilibrium in general-sum normal-form games is known to be PPAD-hard [DGP09, CDT09], so computing a mixed local Nash equilibrium is also PPAD-hard. [DSZ21] indeed considers pure local Nash equilibria, but they show that this is PPAD-hard even in two-player zero-sum games with non-concave utility functions.
>
> We will expand upon these points in the revised version of our paper, providing a clearer bridge between the concepts of pure and mixed strategies and their respective computational implications.
>
> **Q3**: *Does Lemma 1 assume Lipschitz smoothness/continuity on the convex regrets or no?*
>
> **A**: Thanks for this question.  Lemma 1 holds when there exists an online learning algorithm that has $\Phi$-regret against convex loss functions. In the context of lemma 1, we do not need other assumptions. $G$-Lipschitzness is required to achieve sublinear regret for specific algorithms like GD / OG, . But Lemma 1 still holds if there exists a no-$\Phi$-regret algorithm that does not require such assumptions.

---

> ### Author Response · Authors · 2023-11-18
> **Response to Reviewer kZP2-Part 2**
>
> **Q4**: *How does part 2 of Lemma 1 relate to Hazan et al’s [2] results and in general how do the authors’ result relate to your results on projected \phi regret?*
>
> **A**: Firstly, it's important to clarify that Part 2 of Lemma 1 and the results by Hazan et al. [HSZ17] study very different notions of regret (See examples below for an illustration), leading to distinct local equilibrium concepts. As such, they are not directly comparable. Moreover, we've included a detailed discussion in the related works section of Appendix A. The discussion is as follows.
>
> #### The work most closely related to ours is [HSZ17]. The authors propose a notion of *$w$-smoothed local regret* against non-convex losses, and they also define a local equilibrium concept for non-concave games. They use the idea of *smoothing* to average the loss functions in the previous $w$ iterations and design algorithms with optimal $w$-smoothed local regret. The concept of regret they introduce suggests a  local equilibrium concept. However, their local equilibrium concept is very non-standard in that its local stability is not with respect to a distribution over strategy profiles sampled by this equilibrium concept. Moreover, the path to attaining this local equilibrium through decentralized learning dynamics remains unclear. The algorithms provided in [HSZ17] require that every agent $i$ experiences (over several rounds) the average utility function of the previous $w$ iterates, denoted as $F^t_{i,w}:=\frac{1}{w} \sum_{\ell=0}^{w-1} u_i^{t-\ell}(\cdot,x_{-i}^{t-\ell})$. Implementing this imposes a significant coordination burden on the agents. In contrast, we focus on a natural concept of local correlated equilibrium, which is  incomparable  to that of [HSZ17], and we also show that efficient convergence to this concept is achieved via decentralized gradient-based learning dynamics.''
>
> The following two examples shows that the $w$-smoothed local regret and $\Phi_{proj}(\delta)$ (we will refer to it as $\delta$-projected regret) are incomparable. A sequence of actions may suffer linear $w$-smoothed local regret but constant $\delta$-projected regret (Example 1), and vise versa (Example 2).
> **Example 1**:
> 1. Loss sequence: $f^t(x) = |x|$ for every $t \in [T]$ over $X = [-1,1]$.
> 2. Action sequence: $x^t = 1/2$ for odd $t$; $x^t = -1/2$ for even $t$
> 3. Regret: For any $\eta < 1/2$ and $w \ge 1$, the $w$-local regret is at least $\Omega(T / w)$; For $\delta > 0$,  the $\delta$-projected regret of $\{x^t\}$ is at most $\delta$.
>
> **Example 2**:
> 1. Loss sequence: $f^t(x) = -x^2$ for every $t \in [T]$ over $X = [-1,1]$.
> 2. Action sequence: $x^t = 0$ for all $t \in [T]$
> 3. Regret: For any $\eta > 0$ and $w \ge 1$,  the $w$-local regret is $0$; For any $\delta < 1$,  the $\delta$-projected regret of $\{x^t\}$ is at least $\delta^2 T.$
>
> **Q5**: *Naive regret bound in section 3.1 seems meaninglessly loose. That is, having an additive Lipschitz continuity constant G suggests that the algorithm might make no progress at all?*
>
> **A**: The regret bound of naive $\Phi$ regret minimization is $O(\delta G\sqrt{T \log|\Phi^\gamma|} + \gamma G T)$. By setting $\gamma = 1/T$ we can get a regret bound that is sublinear in $T$. However, as we mentioned in the paper, this approach leads to a regret bound that depends on $ \log|\Phi^\gamma|$ while the per-iteration complexity depends on $|\Phi^\gamma|$, which is prohibitively high.
>
> **Q6**: *Reg_proj does not have a learning rate in the step it takes. This seems to affect the notion that projected and external regret can in general be unrelated?*
>
> **A**: Reg_proj (we will refer to it as projected regret) is parameterized by $\delta$, which is the largest $\ell_2$ norm of a possible deviation. As we demonstrated in the paper (example 1 and 2), projected regret is indeed incomparable with external regret in the sense that (1) a sequence of actions could have linear external regret but 0 projected regret; (2) a sequence of actions could have linear projected regret but 0 external regret. Despite the difference, we show that the online gradient descent algorithm achieves sublinear regret bounds for both notions of regret.
>
> [HSZ17]  Hazan, Elad, Karan Singh, and Cyril Zhang. "Efficient regret minimization in non-convex games." ICML, 2017
>
> [DSZ21]  Daskalakis, Constantinos, Stratis Skoulakis, and Manolis Zampetakis. "The complexity of constrained min-max optimization." STOC, 2021
>
> [DGP09]  Daskalakis, Constantinos, Paul W. Goldberg, and Christos H. Papadimitriou. "The complexity of computing a Nash equilibrium." SICOMP, 2009
>
> [CDT09] Chen, Xi, Xiaotie Deng, and Shang-Hua Teng. "Settling the complexity of computing two-player Nash equilibria." JACM, 2009

---

> > ### Comment · Reviewer_kZP2 · 2023-11-20
> >
> > Thank you to the authors for their thorough answers! This is very helpful!
> >
> > I am extremely curious to also hear of a comparison between your local-correlated equilibrium solution concept which is both universal and tractable, and other potential-function based solution concepts (e.g., local minima of the exploitability) which are also universal and tractable. Can you please comment on what would be the advantages vs. disadvantages of computing a local minimum of the exploitability vs. a local correlated equilibrium. See for instance [1] for how a stationary point of the exploitability can be computed in all smooth games.
> >
> > [1] Goktas, Denizalp, and Amy Greenwald. "Exploitability minimization in games and beyond." Advances in Neural Information Processing Systems 35 (2022): 4857-4873.

---

> > > ### Author Response · Authors · 2023-11-21
> > > **Response to Reviewer kZP2**
> > >
> > > **Q**: *I am extremely curious to also hear of a comparison between your local-correlated equilibrium solution concept which is both universal and tractable, and other potential-function based solution concepts (e.g., local minima of the exploitability) which are also universal and tractable. Can you please comment on what would be the advantages vs. disadvantages of computing a local minimum of the exploitability vs. a local correlated equilibrium. See for instance [1] for how a stationary point of the exploitability can be computed in all smooth games.*
> > >
> > > **A**: Thanks for this question! It is a very interesting idea to consider solution concepts based on potential functions in non-concave games. The stationary point of the exploitability [1] represents a universal solution concept. One notable advantage of computing the stationary point of exploitability is that we may be able to apply many off-the-shelf non-convex optimization algorithms to the potential function. In the following, we provide a detailed comparison with our proposed solution concept.
> > >
> > > 1. **It is not clear if we can efficiently compute a stationary point of the exploitability in non-concave games**. We would like to clarify that results of [1] rely on the assumption that each player’s utility function is **concave** with respect to their strategy (page 4 line 3: definition of a pseudo game). Their Theorem 4.1, which enables efficient computation of stationary points (Theorem 4.2), relies on the concave utility assumption (page 20: proof of Theorem 4.1). Therefore, it is not clear whether the stationary point of the exploitability is tractable in non-concave games.
> > > 2. A stationary point of the exploitability, which is a **joint** potential function, does not provide **individual** incentive guarantees. A player at such a stationary point may have strong incentive to deviate.
> > > 3. Optimizing the joint potential function may require heavy communication/coordination among players. It is unclear how decentralized learning dynamics could find such a solution concept. In contrast, our notion of $\Phi$-local correlated equilibrium is closely related to online learning. We show efficient convergence of decentralized learning dynamics to two instantiations of local correlated equilibrium.
> > >
> > >
> > > [1] Goktas, Denizalp, and Amy Greenwald. "Exploitability minimization in games and beyond." Advances in Neural Information Processing Systems 35 (2022): 4857-4873.

---

> > > > ### Comment · Reviewer_kZP2 · 2023-11-22
> > > >
> > > > Thank you this is a great summary. I think that the paper triggering such an extensive discussion amongst the AC and reviewers, whether if the contributions of the paper will be impactful or not, suggests that it would allow the community to reflect and have healthy debates on issues of universal solution concepts and tractability. This, in my opinion, is enough of a contribution for publication on its own. I remain positive on the paper.

---

### Official Review · Reviewer_Ge5S · 2023-11-06

**Soundness:** 3 good
**Presentation:** 2 fair
**Contribution:** 3 good
**Rating:** 6
**Confidence:** 4

**Summary:**

The paper studies the problem of learning equilibria in non-concave (smooth) games. It introduces a new notion of local equilibrium, coined local correlated equilibrium, which is a variation of the correlated equilibrium in which only bounded (local) deviations are allowed. The paper shows that such an equilibrium always exists and it shows that classical no-regret algorithms such as online gradient descent and optimistic gradient efficiently converge to some special cases of such an equilibrium in non-concave (smooth) games.

**Strengths:**

I found the problem studied in the paper really interesting. Understanding which equilibria can be learned efficiently in non-concave games is an important step towards applying game-theoretical solution concepts in modern machine learning problems.

The results presented in the paper are not incredibly complicated from a technical viewpoint, but they nevertheless provide a neat novel analysis of some existing algorithms, shedding the light on what these algorithms actually learn in settings beyond basic games with concave utilities.

**Weaknesses:**

I found that the paper writing is not sufficiently neat in some parts. While all the concepts and results are introduced and adequately explained, there are some issues with terminology and notation, which is not coherent across different sections. For example, in Section 3 the paper talks about differential games, but these have never been introduced in the previous sections (only the definition of smooth game is provided).

My score reflects the weakness above. I strongly encourage the authors to carefully proof read the paper in order to improve it, and I am willing to increase my score if they do so.

**Questions:**

No questions.

---

> ### Author Response · Authors · 2023-11-18
> **Response to Reviewer Ge5S**
>
> **Comment**: *I found that the paper writing is not sufficiently neat in some parts. While all the concepts and results are introduced and adequately explained, there are some issues with terminology and notation, which is not coherent across different sections. For example, in Section 3 the paper talks about differential games, but these have never been introduced in the previous sections (only the definition of smooth game is provided). My score reflects the weakness above. I strongly encourage the authors to carefully proofread the paper in order to improve it, and I am willing to increase my score if they do so.*
>
> **A**: Thank you for your feedback. We acknowledge the issues you've pointed out regarding our writing, particularly in the use of terminology and notation across different sections. We have conducted a thorough review of the paper for clarity and consistency. Specifically, we revised the introduction of differentiable games and smooth games in Section 2, ensuring a more coherent flow into Section 3. The updated version of the paper, with these changes highlighted, has been uploaded for your review.
>
> We appreciate your willingness to reconsider your score upon improvements. We are committed to continuing our proofreading efforts to ensure consistency. If you find any further issues or have additional suggestions in the revised version, please do not hesitate to inform us. Your feedback is invaluable in helping us refine our work.

---

### Comment · Area_Chair_Q1Dc · 2023-11-18
**Some questions to the authors**

Dear authors,

Thank you for your timely responses to the reviewers' comments.

I would first like to ask the reviewers to go through the posted rebuttals and follow up on their concerns or ask for any clarifications before the discussion phase closes.

At the same time, after going through the comments of Reviewer J78b and your replies, I would also like to take this opportunity to ask some questions of my own:

1. First, I would like to understand how to read Theorem 1 (I have similar questions for the other theorems, but Theorem 1 serves just as well for the discussion). In particular, if we take $\delta = D_\mathcal{X}$ and all $f_t$ equal to some fixed non-convex function $f$, Theorem 1 would imply that, after $T$ iterations, OGD attains a value which is $\mathcal{O}(D_\mathcal{X} G /\sqrt{T})$-close to the ***global minimum*** of $f$. Clearly, this cannot be true, as this would mean that non-convex problems could be solved to global optimality in polynomial time. I suspect that you are implicitly assuming that $\delta$ is sufficiently small so that the denominator in the number of iterations required be positive, i.e., $\delta = \mathcal{O}(\sqrt{\epsilon})$ – is this correct?

2. I also have a similar question to that of Reviewer J78b on your concept of local equilibrium. Specifically, consider the single-player maximization game with $\mathcal{X} = [-1,1]$ and $u(x) = x^2/2$: the Nash equilibria of this game are $\pm 1$ (the function's global maximizers), while the critical point of $u$ at $0$ is a global minimizer – that is, the worst possible action choice. However, if we take $\delta \leq \sqrt{2\epsilon}$, the global minimum at $0$ is a $(\delta,\epsilon)$-equilibrium, and hence a $(\Phi_\delta,\epsilon)$-correlated equilibrium as well. Am I missing something?

Kind regards,

The AC

---

> ### Author Response · Authors · 2023-11-20
> **Response to AC Q1Dc**
>
> Thank you for your comment! We address your questions below.
>
> **Q1**: *"First, I would like to understand how to read Theorem 1 ... – is this correct?"*
>
> **A**: This is correct. Our results on convergence to local correlated equilibrium (Theorem 1, 2 and 5) rely on Lemma 1 and hold in the *local regime* where $\delta < \sqrt{2\epsilon / L}$ with $L$ being the smoothness constant. We have included this condition in the statement of Theorem 1, 2 and 5 in the revised paper. We remark that we also provide hardness results for both the computational and oracle complexity of local correlated equilibria in the *global regime* where $\delta = D_{\mathcal{X}}$ (Appendix D).
>
> **Q2**: *"Specifically, consider the single-player maximization game with $\mathcal{X} = [-1,1]$ and $u(x) = x^2/2$ ... Am I missing something?"*
>
> **A**: In this paper, our goal is to introduce a solution concept that is universal, meaning it exists in all games, and is also efficiently computable. In your example, $x=0$ is indeed a local correlated equilibrium based on our definition. But it turns out that including stationary points seems to be the only way to guarantee universality and computational tractability, which we detail in the following paragraph.
>
> **Allowing stationary points is necessary**: The only solution concept previously known to guarantee existence and finite representation — essential for efficient computation — is the local Nash equilibrium. However, it has been demonstrated to be computationally intractable in the local regime in [DSZ21]. Therefore, we propose a solution concept that relaxes the local Nash equilibrium and is aditionally efficiently computable. To ensure existence, the set of local Nash equilibria must include stationary points. This can be illustrated with a basic zero-sum (min-max) game, characterized by the utility function $u(x,y) = (x - y)^2$, where $x, y \in [-1, 1]$. In this game, a simple analysis reveals that no global Nash equilibrium exists. Likewise, there isn't a solution where the min-player chooses a local minimum and the max-player a local maximum. However, any point $(x, y)$ with $x = y$ is a local Nash equilibrium, as at these points each player encounters a zero gradient, indicating no incentive to deviate when considering only local gradient information. Our concept of local correlated equilibrium, being a more generalized version of local Nash equilibrium (encompassing it as a special case), naturally includes stationary points. The inclusion of stationary points is an unavoidable compromise; excluding them would lead to the loss of either universality or finite representation in the solution concept.
>
> [DSZ21] Daskalakis, Constantinos, Stratis Skoulakis, and Manolis Zampetakis. "The complexity of constrained min-max optimization." STOC, 2021

---

> > ### Comment · Reviewer_kZP2 · 2023-11-20
> >
> > Thanks to the AC for raising some interesting questions!
> >
> > Dear authors, I hope I am not missing something obvious but in the example you give $u(x,y)\doteq (x-y)^2$ how is $x=y$ a local Nash? it seems like $y$ can always $\delta$ deviate to increase it's payoff since it can always move away from x?

---

> > > ### Comment · Reviewer_J78b · 2023-11-20
> > >
> > > If I'm not mistaken the definition of local Nash equilibrium is wrt to first order deviations. So in the game $u(x,y) = (x-y)^2$, deviating by $\delta$ from $x=y$ only has $\delta^$ magnitude so it doesn't violate the definition of local Nash.

---

> > ### Comment · Area_Chair_Q1Dc · 2023-11-20
> >
> > Thank you for your reply.
> >
> > My question had to do with whether the proposed notion of a local equilibrium is "game-theoretically meaningful" (as you state in the introduction) or not. Given that this solution concept includes the *worst* possible action that a player can take in a single-player game, I do not see how it is in any way rationally justifiable.
> >
> > It is also clear that there are non-concave games that do not admit local Nash equilibria. This brings us back to the GAN training example of Reviewer J78b: in that case, the players' learning trajectories will not converge to anything (they may cycle or exhibit any other type of non-convergent behavior), so proposing a certain probability measure on the players' action spaces as the "outcome" of their interactions is not informative.
> >
> > In short, regarding the question of [Das22] "*What solution concepts are meaningful, universal, and tractable?*", the fact that a solution concept is "universal" and "tractable" does not imply that it is also "meaningful".
> >
> > Regards,
> >
> > The AC

---

> > > ### Comment · Reviewer_kZP2 · 2023-11-20
> > >
> > > Thank you Reviewer J78b for the clarification!
> > >
> > > I should really let the authors answer first but I would like to also chip-in with the following in case if some of the reasons why I liked the paper might be misguided. I think that there are a few odd aspects to the solution concept proposed and the content of the paper (which the AC as well as reviewer J78b rightfully pointed out), that said I think the solution concept still seems interesting for the following two reasons(, and please correct me if anything in my reasoning is wrong!):
> > >
> > > 1) In finite state and action Markov games, the computation of CCE (in stationary policies) is intractable [1], this work seems to provide the first tractable solution concept for such games in stationary policies (albeit I am not sure at first sight if this local CCE is locally optimal if players can locally best-respond with non-stationary policies---which might be an issue).
> > >
> > > 2) The paper seems to almost suggest a new paradigm for the training and evaluation of "multiagent" neural network, e.g., GANs (btw, I think that GANs are imo not a great example for this multiagent setting because in GANs we do not really care about the final performance of the discriminator, we only about the final performance of the generator, a better example are neural networks with a shared set of weights with individual training objectives). Traditionally, in deep learning, we take the last or best iterate of training, and use these weights to evaluate our network for all points in our test set. The author's solution concept suggests an alternative evaluation paradigm (albeit potentially memory intensive but this could be imo remedied with additional research). Namely, one could store a subset of the weights encountered during training, and then at test time for each data point in the test set evaluate the networks using a randomly sampled weights from the subset of stored weights. By the definition of the local-CCE (I am ignoring here the deviation set chosen), then on average the payoffs achieved by the neural networks should be higher on the test set than if we chose to only evaluate according to the last/best iterate.
> > >
> > > **References**
> > >
> > > [1] Daskalakis, Constantinos, Noah Golowich, and Kaiqing Zhang. "The complexity of markov equilibrium in stochastic games." The Thirty Sixth Annual Conference on Learning Theory. PMLR, 2023.

---

> > > ### Author Response · Authors · 2023-11-21
> > > **Response to Area Chair Q1Dc**
> > >
> > > **Q1**: *My question had to do with whether the proposed notion of a local equilibrium is "game-theoretically meaningful" (as you state in the introduction) or not. Given that this solution concept includes the worst possible action that a player can take in a single-player game, I do not see how it is in any way rationally justifiable.*
> > >
> > > **A**: The proposed solution concept of local correlated equilibrium provides an incentive guarantee that no player can use first-order gradient information to achieve large local improvement. Ideally, one would like every player to be at a local maximum of their utility function. As we pointed out in our example, i.e., $\\min_{x \in [-1,1]} \\max_{y \in [-1,1]} (x-y)^2$, such a solution might not exist. Additionally, even recognizing whether a given point is a local maximum is NP-hard [MK87], so it is difficult for a computationally bounded player to verify. To circumvent this difficulty, we decide to allow players to only consider simpler types of deviations, e.g., local deviations based on only first-order information. One might consider allowing higher-order information. This has two drawbacks: (i) it might destroy the existence even if we only consider second-order information (see the example), and (ii) even when the local equilibrium exists, we face the same issue, namely, it might still be the worst possible action. Just imagine a player facing $u(x) = x^{100}$ at the worst action $x = 0$ does not have incentive to deviate with up to the 99th-order information. Finally, note that stationary points have been considered as a game theoretic relevant solution concept in the literature, e.g., [HSZ17].
> > >
> > > **Q2**: *It is also clear that there are non-concave games that do not admit local Nash equilibria. This brings us back to the GAN training example of Reviewer J78b: in that case, the players' learning trajectories will not converge to anything (they may cycle or exhibit any other type of non-convergent behavior), so proposing a certain probability measure on the players' action spaces as the "outcome" of their interactions is not informative. In short, regarding the question of [Das22] "What solution concepts are meaningful, universal, and tractable?", the fact that a solution concept is "universal" and "tractable" does not imply that it is also "meaningful".*
> > >
> > > **A**: We have made it clear in the response to Review j78b that *“it’s important to note that training GANs is a min-max optimization problem where the goal is to identify the minimax strategy for the min-player (controlling the generator network). Our focus on finding a local correlated equilibrium is very different from the pursuit of a minimax strategy, and as such, doesn’t inherently guarantee high performance of the generator.”* As also highlighted by  Review j78b, (coarse) correlated equilibrium is a useful concept when there is a signaling device that could be used to correlate different agents’ actions. Our solution concept, $(\epsilon, \Phi_{int}(\delta))$-local correlated equilibrium when applied to concave games, corresponds exactly to the notion of coarse correlated equilibrium. Unfortunately, in the context of non-concave games, finding a (coarse) correlated equilibrium is computationally intractable. We believe our solution concept serves as a useful alternative in these non-concave games where a signaling device is available, which includes many multi-agent systems such as market dynamics, cooperative strategies among self-driving cars, and interactions within social networks. Finally, our new solution concept is meaningful because (1) it provides stability for boundedly rational agents (see our answer to Q1), (2) it corresponds to the well-studied game theoretic concept of coarse correlated equilibrium in concave games, and (3) it includes local Nash as a special case in non-concave games.
> > >
> > > [MK87]  Murty, Katta G., and Santosh N. Kabadi. "Some NP-complete problems in quadratic and nonlinear programming." Mathematical Programming: Series A and B, 1987
> > >
> > > [HSZ17] Hazan, Elad, Karan Singh, and Cyril Zhang. "Efficient regret minimization in non-convex games." ICML, 2017

---

> > > > ### Comment · Area_Chair_Q1Dc · 2023-11-22
> > > >
> > > > Thank you for your continued input.
> > > >
> > > > From your replies, it seems that you are again primarily motivated by tractability - and while first-order information is of course insufficient *at* a stationary point, it points away from this point from any other point in your $x^100$ example (and, in fact, even a zeroth-order comparison with a nearby point is sufficient to detect this).
> > > >
> > > > In short, a point cannot be rationally admissible if there are nearby deviations yielding a strictly greater payoff: this is the most basic tenet of rationality, and the proposed solution concept does not satisfy it.
> > > >
> > > > At any rate, after the last round of replies, I have all the information needed for the subsequent discussion phases. Thank you again for your timely input.
> > > >
> > > > Regards,
> > > >
> > > > The AC

---

> > > > > ### Comment · Reviewer_J78b · 2023-11-22
> > > > > **A point *can* be rationally admissible if there are nearby deviations yielding a negligibly greater payoff**
> > > > >
> > > > > Hi AC,
> > > > > Sorry- I'm just a reviewer here, but I strongly disagree about the idea that in order to be "rational" it is absolutely necessary to not have any, even negligible, deviations.
> > > > >
> > > > > For a simple real life example, when reviewing papers to a conference we have a slight incentive to bring down their scores because if they get rejected that leaves slightly more room in the conference for our papers. But we don't do it- we review the papers as honestly as we can because that's the right thing to do. On the other hand if your students' paper was contending with a paper you're reviewing for a single slot, then your incentive would be very strong and you might be tempted to push your students' paper.
> > > > >
> > > > > I don't completely buy the motivation for the new solution concept (although the discussion has improved my opinion significantly), but I think the first-order condition is pretty natural.
> > > > > Best,
> > > > >
> > > > > PS: Also, if you're really concerned about the first-order issue, I beleive that all the complexity hardness results hold for multi-linear games (which come up naturally in game theory).

---

> > > > > > ### Comment · Reviewer_kZP2 · 2023-11-22
> > > > > >
> > > > > > I actually think the AC's point is valid and in some sense suggests that deviations in terms of the projected gradient operator is not a great set of deviations (even though it is an obvious one in machine learning). I think the important question to ask as a follow-up to this paper is what are the set of deviations that are meaningful in non-convex settings. I am personally not convinced of the two sets of deviations proposed in this paper but I think it is a starting point. Answering this question should be in many ways a debate which will consider questions of "local rationality" and could allow us to derive algorithms which then converge to better solutions.

---

> > > > > > ### Comment · Area_Chair_Q1Dc · 2023-11-22
> > > > > >
> > > > > > Dear Reviewer J78b,
> > > > > >
> > > > > > First of all, let me say that no one is "just a reviewer" or "just an AC" or "just an author" :-) Everything here is evaluated on scientific merit, and no one is infallible - we're all human, authors, reviewers, ACs, SACs, and PCs may be as mistaken or wrong as any other, so your input is greatly valued and I'm sincerely obliged for your engagement (as with everyone else's).
> > > > > >
> > > > > > Now, on to your main point: there is a very big difference between games with individually concave payoff functions (such as multilinear games) and non-concave games (like the authors are considering).
> > > > > >
> > > > > > In multilinear games - and, more generally, in all games with individually concave (or even pseudo-concave) payoff functions - first-order stationarity *implies* that the point in question is a Nash equilibrium. This is one of the main reasons that the notion of $\epsilon$-equilibrium is the "gold standard" of approximate equilibrium in multilinear / concave games (and why, for instance, there are no rationality concerns regarding the membership of $\epsilon$-equilibria of **finite** games in PPAD in the famous result of Daskalakis et al.).
> > > > > >
> > > > > > Put differently, when the underlying game is multilinear / concave / pseudo-concave, one can freely consider only first-order conditions and notions, and be done with it: a point which is first-order stationary is fully justifiable in this case.
> > > > > >
> > > > > > By contrast, when the underlying game is *not* concave, first-order stationarity can mean playing a globally dominated strategy (again, think of maximizing $u(x) = x^2/2$ over $[-1,1]$), and this is not consistent with the von Neumann - Morgenstern axioms of utility theory.
> > > > > >
> > > > > > I hope this clarifies my objection to the authors' claim in the introduction (and in the discussion so far) that the proposed notion of equilibrium is "game-theoretically meaningful".
> > > > > >
> > > > > > Regards,
> > > > > >
> > > > > > The AC

---

### Author Response · Authors · 2023-11-23
**Summary of our motivation, results and general thoughts on meaningful equilibrium concepts (Part 1)**

We thank all reviewers and the area chair for their discussion, and constructive comments that significantly improve the presentation of the paper, and provide some extra points for us to discuss in the introduction. We are encouraged by the recognition of our work's merits and the fact that there is good debate about our paper’s contributions. For an equilibrium concept paper, debates such as these are natural to expect and encouraging. Below we provide a summary of our motivation, results and general thoughts on what would be a meaningful equilibrium concept in non-concave games, addressing some of the natural critiques of the AC and some of the reviewers.

In sum, our paper proposes a solution concept that is universal, tractable and, we believe, is game-theoretically meaningful. In particular, we propose a relaxation of the first-order local Nash equilibrium of Daskalakis-Skoulakis-Zampetakis, which has the added bonus of being tractable. We are not aware of any other solution concept that achieves all these properties simultaneously.

We do find our proposed solution concept is meaningful for the following reasons:

(1) it offers stability wrt local deviations for boundedly rational agents who only have a limited understanding of their utility; namely, for agents who only have access to zero-order/first-order information about their utilities

(2) it corresponds to the well-studied game-theoretic concept of coarse correlated equilibrium in concave games, and

(3) it contains first-order local Nash equilibrium as a special case in non-concave games.

Naturally, one could critique our solution concept and ask:

i. “why not modify our solution concept and insist that every player plays a global optimum?”

ii. “why not modify our solution concept and insist that every player plays a local optimum?”

iii. “why not modify our solution concept and insist that every player plays a second-order local optimum?”

The short answer is that any of i, ii, or iii are not an option. Namely, if we insisted on any of these, the resulting equilibrium concept would either not exist or have infinite description complexity. Besides these issues, there is an additional one. While it is easy to check that 0 is a local maximum of $-x^2$ and 0 is not a local maximum for $x^2$, in general it is NP-hard to check if a given point is a local maximum of a function. This poses an additional challenge that does not have to do with existence/tractability of the solution concept: if we insisted on ii (or i above) it would be NP-hard for a player to tell that they are at equilibrium, and this would not be meaningful if they agents are computationally bounded (as they are in the real world).

In view of all the above, we are making the best compromise we can think of. We assume that the agents are boundedly rational: they have 0-th and 1-st order access to their utilities, and use this information to verify whether they can gain from local deviations. Assuming 0-th and 1-st order access to utilities is reasonable for the high-dimensional/deep learning settings that motivate our study. Higher-order access to the utilities would a. be too expensive; b. motivate notions of equilibrium that do not exist/have infinite description complexity, even at the second level of the hierarchy (namely second-order local equilibria); and c. if we are too greedy (insisting on local max equilibria) not only do we lose existence but also get intractability of verifying that we are at equilibrium.

With this choice, we propose a solution concept that is universal and tractable, and guarantees first-order stability which implies a deviation of $\delta$ gives at most second-order $O(\delta^2)$ improvement. Regarding the question of approximation, we emphasize that approximate Nash/correlated/other equilibria are widely used in Game Theory, Online Learning, Computer Science, other theoretical or applied fields. One may complain "why not grab an extra utility of epsilon, however small, if there is a way to deviate to gain that extra epsilon?" The reason is that: (i) it may be costly to implement strategy changes, so the player might be willing to ignore gains that are relatively minor compared to the required changes (e.g., $\delta^2$ gain vs. $\delta$ change); (ii) a player must really believe in their utility functions’ accuracy to think this epsilon potential is real— in many cases the utility is itself estimated from data so there is already an approximation error in the utility.

---

> ### Author Response · Authors · 2023-11-23
> **Summary of our motivation, results and general thoughts on meaningful equilibrium concepts (Part 2)**
>
> It goes without saying that no equilibrium concept is perfect. Even for well-established ones, like Nash equilibrium, there is a vast literature giving example settings where we don’t like its predictions, proposing refinements or other solution concepts to fix these problems, and this keeps going. In our setting of non-concave games, by no means do we want to suggest that our proposed solution concept solves the problem of identifying the right equilibrium target. We think that this is an extremely hard problem, but one that is important already and will be even more important in the future. Unfortunately, prior to our work, there had just not been a proposed solution concept that exists and is tractable and that is (even very weakly) stable under individual player deviations. We are happy with the progress that is made in our paper and it is an invitation to the community to think about this important problem. Game-theoretic settings are extremely hard, and there cannot be progress if small steps are quashed.
>
> Just to give some historical context here, when Nash went to one of the most prominent mathematicians of his time (von Neumann) to present his theorem, von Neumann dismissed it as trivial. Anecdotally, it is believed that von Neumann said something like “you know this is just a fixed point.” von Neumann is right. Nash equilibrium is a very weak solution for many reasons, much weaker than the much sharper minimax equilibrium that von Neumann had worked on. However, Nash’s theorem has been much more impactful than von Neumann’s because, despite its weaknesses, Nash equilibrium is guaranteed to exist beyond two-player zero-sum games. We similarly invite you not to dismiss our contribution. This is truly the best solution concept we can guarantee exists, is tractable and satisfies some weak unilateral deviation property. It is really an invitation to think along this plane (of existence, tractability, meaningfulness) for better solution concepts.

---

### Meta-Review · Area_Chair_Q1Dc · 2023-12-06

**Metareview:**

This paper introduces and examines the notion of a *local correlated equilibrium* for $N$-player games with continuous action spaces and non-concave utility functions. This notion is expressed in terms of probability distributions over the players' action sets, and it is defined similarly to how coarse correlated equilibria are defined in finite games – but with only local deviations allowed (hence the name). The authors connect this notion to certain instances of $\Phi$-regret minimization, and they also connect the statistical distribution of states visited by gradient algorithms to the proposed solution concept.

This paper generated a very rich, diverse, and extensive discussion at all stages of the reviewing process. On the positive side, the more positively-inclined reviewers appreciated the result that, in any smooth game, gradient algorithms converge to some solution concept, even when the players' payoff functions are not concave. On the more negative side, there was considerable debate as to whether (i) the mode of convergence is sufficiently expressive; and (ii) the proposed solution concept is sufficiently meaningful.

One concern which remained after several exchanges with the authors had to do with the fact that, in non-concave games, there is a fundamental trade-off between tractability and meaningfulness which does not arise in finite / concave games. Specifically, in finite / concave games, (approximate) stationarity implies (approximate) equilibrium; in non-concave games it does not, and a global utility minimizer would still satisfy the authors' proposed definition of a "local correlated equilibrium". In other words, the proposed notion does not capture "approximate equilibria" but "approximate stationary points", a distinction which is absent from the paper, and which leads to an unbalanced presentation (which should be avoided in a paper introducing a new solution concept).

A further issue concerned the focus on the notion of $\Phi$-regret. This notion is very general and, indeed, it encompasses both external and swap regret. However, the paper only treats a specific, "interpolated" instance of $\Phi$-regret (and a projected variant thereof), which only allows small, fixed deviations from the players' actual strategies. Since this is quite far from the more standard notions of external and swap regret, the special case of $\Phi$-regret considered in the paper should also be justified – after all, not all subclasses of $\Phi$-regret are meaningful.

Finally, one last issue that was discussed extensively with the committee during the last phase had to do with the fact that the proposed solution concept does not concern the players' actions, but probability distributions over the players' joint action profiles. There are well-known examples in the literature where GD/OGD converge to a limit cycle from any initialization, and the trajectories' time-averages converge to a point which is not even a stationary point  – cf. the “Polar Game” of Pethick et al. (ICLR 2022, https://arxiv.org/pdf/2302.09831.pdf) or the “Forsaken Matching Pennies” example of Hsieh et al. (ICML 2021, https://arxiv.org/abs/2006.09065). The authors' analysis does not contradict these examples because it concerns coarser statistical / distributional properties of GD/OGD that are not reflected even in the time-averaged trajectories of the algorithms; however, the existence of these examples also goes to show that the type of convergence considered is not sufficiently expressive to capture the true behavior of GD/OGD in non-concave games, something which should also be discussed at length.

Summing up, even though this paper contains interesting ideas, the current treatment is not yet ready for publication, and an extensive revision and fresh round of reviews would be required before considering an "accept" recommendation. For this reason, a decision was reached to make a "reject" recommendation at this stage while encouraging the authors to review and enrich their paper and resubmit.

**Justification For Why Not Higher Score:**

Important limitations were not discussed in the paper.

**Justification For Why Not Lower Score:**

N/A

---

### Decision · Program_Chairs · 2024-01-16

Reject